# Human Vitamin K Epoxide Reductase as a Target of Its Redox Protein

**DOI:** 10.3390/ijms23073899

**Published:** 2022-03-31

**Authors:** Julie Ledoux, Maxim Stolyarchuk, Enki Bachelier, Alain Trouvé, Luba Tchertanov

**Affiliations:** Centre Borelli, ENS Paris-Saclay, CNRS, Université Paris-Saclay, 4 Avenue Des Sciences, F-91190 Gif-sur-Yvette, France; julie.ledoux@ens-paris-saclay.fr (J.L.); maksim.stoliarchuk@ens-paris-saclay.fr (M.S.); enki.bachelier@universite-paris-saclay.fr (E.B.); alain.trouve@ens-paris-saclay.fr (A.T.)

**Keywords:** human vitamin K epoxide reductase, hVKORC1, blood coagulation, redox protein target, enzyme activation, modular protein, intrinsically disordered L-loop, conformational transition, modelling, molecular dynamics simulations, free energy landscape

## Abstract

Human vitamin K epoxide reductase (hVKORC1) enzymatic activity requires an initial activation by a specific redox protein, a less studied step in the hVKORC1 vital cycle. Significant steric conditions must be met by enzymes, being that to adapt their configurations is mandatory for hVKORC1 activation. We studied, by molecular dynamics (MD) simulations, the folding and conformational plasticity of hVKORC1 in its inactive (fully oxidised) state using available structures, crystallographic and from de novo modelling. According to the obtained results, hVKORC1 is a modular protein composed of the stable transmembrane domain (TMD) and intrinsically disordered luminal (L) loop, possessing the great plasticity/adaptability required to perform various steps of the activation process. The docking (HADDOCK) of Protein Disulfide Isomerase (PDI) onto different hVKORC1 conformations clearly indicated that the most interpretable solutions were found on the target closed L-loop form, a prevalent conformation of hVKORC1’s oxidised state. We also suggest that the cleaved L-loop is an appropriate entity to study hVKORC1 recognition/activation by its redox protein. Additionally, the application of hVKORC1 (membrane protein) in aqueous solution is likely to prove to be very useful in practice in either in silico studies or in vitro experiments.

## 1. Introduction

Vitamin K (K from *koagulation*, German), a natural K-vitamer generated by plants and preferentially transported to the human liver [1], is required for the post-synthesis modification of proteins involved in blood coagulation (blood pro- and anti-clotting enzymes, e.g., endothelial anticoagulant protein S) as well as proteins outside the coagulation cascade [2]. Recently, it was reported that vitamin K levels appear to be extremely reduced in the lungs of hospitalised COVID-19 patients and particularly in those who need mechanical ventilation in the intensive care unit and/or died [3]. It seems, therefore, that the activation of endothelial protein S in these patients is more severely compromised than the activation of hepatic procoagulant factors [4] and is compatible with enhanced thrombogenicity in COVID-19 [5].

Within cells, vitamin K participates in a cyclic process initiated in the endoplasmic reticulum (ER) lumen through a reduction from the inactive vitamin K 2,3-epoxide to the active vitamin K quinone by the enzyme vitamin K epoxide reductase (VKOR) [6,7]. After each vitamin K-dependent protein activation, vitamin K is recycled to the initial inactive state.

Although different hypotheses based on the biochemical, biophysical, in silico, and crystallographic studies have been proposed, explicit mechanisms of vitamin K reduction in mammals, in particular human VKOR (hVKORC1), are not currently well defined. Each suggested mechanism postulates that redox proteins transfer electrons to two conserved cysteines in a luminal loop (L-loop) of hVKORC1. Further, these electrons are transferred to a CXXC motif in the enzyme transmembrane domain (TMD) [8]. Finally, the reduced CXXC motif of hVKORC1 transfers electrons to vitamin K [7,9]. Each step of vitamin K reduction is tightly coupled to the motif CXXC oxidation in the hVKORC1 active site. To repeatedly reduce vitamin K, hVKORC1 must be regularly activated by a redox partner delivering reducing equivalents through a thiol–disulfide exchange reaction. The cooperation of hVKORC1 with a redox protein implies an activation process that represents a less-studied step in hVKORC1’s vital cycle.

Thiol–disulfide exchange reactions are central to oxidative protein folding and a key mechanism in almost all enzymes generating and isomerizing disulfide bonds [10]. Understanding the mechanisms of thiol–disulfide exchange still remains a significant intellectual challenge 50 years after the classic studies of Anfinsen and colleagues on the refolding of reduced ribonuclease A (RNase) [11]. The reaction is initiated by the nucleophilic attack of a thiolate on a disulfide (Figure 1a). An attacking thiolate approaches along the disulfide axis, and this requirement for collinearity establishes the orientation necessary for interactions between well-structured redox partners [12,13]. Thus, disulfide exchange reactions have significant steric requirements that must be met by enzymes capable of adapting their folding at each step of the process.

The study of VKOR is pertinent, as VKOR has dual interests—as a clinical target for the development of vitamin K reduction modulators and as an enzyme activated by its redox protein through thiol–disulfide exchange reactions, a biologically fundamental process. In the present work, we analysed the structural and conformational properties of hVKORC1 to provide an accurate target model for fundamental research and pharmacological applications, focusing on the exploration of hVKORC1 ability to recognise its redox protein.

Recently, by comparing four thioredoxin proteins as promising redox partners of VKOR, we (i) suggested that human Protein Disulfide Isomerase (PDI) is the most probable hVKORC1 redox protein and (ii) proposed molecular complex PDI–hVKORC1 as a first precursor to probe thiol–disulfide exchange reactions between PDI and hVKORC1 [15]. For the complex modelling, the de novo model of hVKORC1 in the oxidised inactive state was used as a target of PDI. The de novo model is composed of the transmembrane four-helix bundle crowned by the intrinsically disordered (ID) luminal loop (L-loop, R33-N77) (Figure 1b), protruding in the endoplasmic reticulum lumen [14].

To advance our knowledge on hVKORC1 as a target for its redox protein, in the present paper we explored all available structural information on the de novo model and recently reported crystallographic structures of VKOR [16].

As we focused on a fully oxidised-state enzyme, in which two pairs of cysteine residues form disulfide bridges, we concentrated our analysis on structures that reported two forms of the oxidised state, which were obtained at different crystallisation conditions and referenced in [16] as open and closed (Figure 1c). These terms are not related to L-loop conformation and describe the forms obtained either by the co-crystallisation of VKOR with ligands, the holo form, or ligand-free enzymes, the apo form. Since crystallographic data revealed structurally different forms of the same enzymatic state with a common topology/connectivity, it perfectly confirms the large flexibility of the L-loop suggested in silico [14].

Focusing on hVKORC1 as the target of its redox protein PDI, we first address the central question—is the hVKORC1 L-loop a properly folded two-state domain that provides a reversible open-to-closed conformational transition (on-off switch), postulated in [16], or is it an inherently disordered (ID) region possessing the great plasticity/adaptability required to perform various steps during the hVKORC1 activation process? Secondly, we examined if hVKORC1 can be considered as a modular protein composed of two-subdomains, the transmembrane domain and the L-loop, having their distinct structural properties and fulfilling their own functions? Finally, in the context of enzyme activation, we asked, what conformation of the hVKORC1 L-loop is an authentic target for the redox protein PDI that may be used for automatic docking experiments?

To explore these multiple questions as much as possible in a more explicit and pragmatic way, we characterised structural and conformational properties of hVKORC1 in the fully oxidised (inactive) state using two crystallographic structures (apo and holo forms) and conformational spaces generated by molecular dynamics (MD) simulations of their homology models, a de novo model with an intrinsically disordered L-loop [14], and the L-loop extracted from these models and simulated as an isolated polypeptide.

We first postulated that two crystallographic forms must be easily reversible during MD simulations under identical conditions (unbound protein in water solution). MD conformational sets derived from two crystallographic forms and de novo models were compared to reach an expected consensus on their common conformational spaces. Second, conformations derived from crystallographic structures and de novo models were used as PDI targets in protein–protein docking trials.

This computational study did not probe the redox sensitivity of hVKORC1 but only comprehensively investigated the structural and conformational properties of enzymes and the estimated binding capacity of different conformations of hVKORC1 with PDI.

## 2. Results

### 2.1. Two Crystallographic Forms of VKOR Fully Oxidised State: Analyses and Hypothesis

This paragraph summarises a comparative analysis of VKOR crystallographic structures recently reported in [16]. This analysis was carried out to improve our knowledge of hVKORC1 regarded as a target of its redox protein, an aspect not yet discussed to date at a structural level.

The crystallographic structures represent two alternative forms of VKOR’s oxidised state obtained upon different crystallisation conditions: by the co-crystallisation of human VKORC1 with vitamin K antagonists (VKAs) and additive molecules (glycerol monooleate), located in the active site pocket, and by the ligand-free VKOR-like protein from *Takifugu rubripes* and ligand-bound VKOR, in which the vitamin K quinone (KQ) or vitamin K epoxide (KO) is located in a non-catalytic site. These two forms were called by the authors open and closed, respectively, for the ligand - free and ligand-bound active site forms. To compare these two crystallographic forms, the respective structures were retrieved from the Protein Data Bank (PDB) [17], and their original atomic coordinates were extracted from well-resolved structures. To avoid inaccuracy in the used terminology for various conformations of VKOR, the forms identified by crystallography will be referred to as apo-c (PDB ID: 6wvi) and holo-c form (PDB ID: 6wv3), while the MD conformations of hVKORC1 with varying L-loop conformations from compact globular-like to elongated arrangements will be referred to as open and closed.

The Root Mean Square Deviation (RMSD) calculated on Cα-atoms (the N- and C-terminal residues were excluded) showed that the two forms differ mainly in the L-loop (4.0 Å), while in the TMD they are similar (0.7 Å) (Figure 1 and Figure 2). The L-loop showed a helical folding composed of two helices (H1-L and H3-L) in apo-c form and three helices (H2-L, H3-L, and H4-L) in holo-c form. Noteworthy, in both forms, is that H3-L is made up of two adjacent small 3_10_-helices separated by R61 apparently acting as a breaker of regular structures. The double 3_10_-helix is formed by the same residues in both structures, and its position in 3D space is equivalent.

In addition to folding differences, the L-loop shows divergent conformations, resulting in a distinct localisation of the L-loop disulfide bridge compared to the one in the active site, where they are distant in apo-c and neighbouring in holo-c.

We first hypothesised that ligands may play a decisive role in stabilising the holo-c form. Surprisingly, there are no H-bonds between ligands and residues of the L-loop. Both ligands, warfarin and glycerol monooleate, form non-covalent interactions (H-bonds and hydrophobic contacts) only with residues from the TM1, TM2, and TM4 of TMD, with the exception of a unique hydrophobic contact described as the interaction C-H···π, between V54 (H2-L) and warfarin (Figure 2b). Therefore, if L-loop residues do not make a significant contribution to ligands binding, the structural and conformational difference of the L-loop in two crystallographic forms depends on other factors. To identify such factors, non-covalent contacts stabilising each crystallographic form were calculated for two sub-regions of the L-loop, defined in [16] as cap (R33-F55) and anchor (S56-N77); the second sub-region, a very flexible in de novo model, will be further called the hinge. Both sub-regions are stabilised by multiple H-bonds differing greatly between two crystallographic forms in the cap (so-called the form-specific contacts), while most H-bonds (five of the nine) in the hinge are observed in both forms (Figure 2d; Appendix A). In general, distances characterising the detected H-bonds in two forms are shorter in apo-c form compared to holo-c.

Remarkably, residue N80 acts as a trifurcated binding centre in both crystallographic forms to maintain H2-L (holo-c) or coil (apo-c) and H3-L at proximity to transmembrane helix TM2 through simultaneous interactions with W59, G60, and F63. Additionally, N80 provides affinity to warfarin as a donor in H-bonds (in holo-c form). Similarly, residue Q78 is involved in bifurcated H-bonding with G62 and I75 in both forms. The form-specific weak H-bonds and hydrophobic contacts involving either intra-L-loop residues alone or with TMD contribute to the incremental stabilisation of L-loop conformation in each form.

This analysis of crystallographic structures showed that (i) the L-loop of VKORC1 in the fully oxidised state is potentially able to display large structural and conformational rearrangements; (ii) L-loop residues do not contribute significantly to ligands binding; (iii) the L-loop in both crystallographic forms is mainly stabilised by form-specific H-bonds in the cap, while the hinge is stabilised by H-bonds common in both forms. We naturally suggested that (i) the L-loop structural and conformational difference in holo-c form versus apo-c is driven primarily by steric factors associated with the accommodation of ligands or their absence, and (ii) removing the ligands from VKOR would relax the target, promoting the closed-to-open transition referred to in [16].

### 2.2. Why Are Two Crystallographic Forms of the Same VKOR State Not Reversible during MD Simulations?

The structure of human VKORC1 crystallised in holo-c form (6wv3) was studied by conventional molecular dynamics (cMD) simulations (200-ns trajectory) without ligands. Surprisingly, the removal of ligands did not produce the expected effects on cMD VKORC1 conformations (Figure 3). Indeed, although cMD data revealed transient folding and the flexibility of the L-loop, evidenced by (i) increased RMSD values; (ii) the instability of the regular fold showing the reversible transition of helices to coil; and (iii) the flexibility of L-loop inter-helices linkers, the observed effects did not provide a significant plasticity, which would lead the transition of holo-c to apo-c form. Upon cMD simulation, the cap H-bonds are significantly weakened or vanished, while almost all H-bonds in hinge are maintained despite highly transient folding.

We hypothesised that (i) the 200-ns cMD simulation was insufficient to observe the expected holo-c to apo-c transition, and (ii) the extended conformational sampling of both forms would help efficiently detect their conformational spaces overlap and observation of the expected transition. To generate comparable datasets for two forms, human full-length VKORC1 homology models, apo-h and holo-h, were built using the crystallographic structures, apo-c and holo-c, as templates. Then, to improve conformational sampling, each form of the human protein was investigated using the robust Gaussian accelerated MD (GaMD) methodology [18].

The analysis of the 500 ns GaMD simulation data (RMSD, RMSF, and secondary structures) showed that the well-conserved TM helices of hVKORC1 vary slightly only at their ends, while L-loop helices exhibit an unstable fold in each form and a tendency to equalise the fold between two forms (Figure 4). Segment P37-C43, either folded as an α-helix (H1-L) in apo-h or as a random coil in holo-h, preserves its structure observed in the crystallographic template. In contrast, other L-loop segments show an unstable fold in both forms. In particular, fragment S52-F55, a coil in apo-c, appears as a transient 3_10_-helix, further stabilised as an α-helix, and corresponds well to H2-L helix observed in holo-h (and holo-c) form. Moreover, segment V66-V69, stabilised as H4-L helix in holo-c, is essentially unfolded in the second half of the GaMD simulation, similar to apo-c. Despite these obvious changes in the L-loop secondary structures, the spatial positions of the cap and hinge regions relative to TMD are preserved in both forms.

Curious fixedness of the L-loop observed during GaMD simulations of models derived from the crystallographic structures of apo and holo forms evokes the question of the L-loop’s role in hVKORC1 activation. L-loop rigidity conflicts with its primary functions—the recognition and binding of redox protein by VKOR—for which the L-loop must be easily adaptable and, therefore, highly flexible. Moreover, the secondary structures variation observed in the L-loop, particularly transitions from the helix to coil or vice versa, leads to the coil elongation associated with the increase of L-loop flexibility.

To understand the factors contributing to L-loop fixedness, we analysed the H-bonds stabilising each form. This analysis revealed that residue Q78 participates in a unique stable H-bond, Q78···G62, were well preserved during GaMD simulations in both forms (occurrence of 100%), while their bonding to I75 was maintained in only 25 and 24% of conformations of apo-h and holo-h forms, respectively (Figure 4f). The trifurcated H-bonding of N80, observed in the crystallographic structures, fully disappeared in both forms during GaMD simulations. Consequently, there is no stable H-bond interaction between the L-loop and TMD that would hold them together in both forms. Possibly, the intra-L-loop Q78···G62 H-bond stabilises the flattened conformation of the hinge and holds this rigidified hinge close to TM2, limiting its displacement.

### 2.3. Role of H-Bond Q78···G62 on L-Loop Inherent Dynamics

We assumed that (i) breaking the Q78···G62 H-bond would give the L-loop more mobility, and (ii) if this contact is functionally crucial for the L-loop, it will be restored upon MD simulation. To examine this hypothesis, two datasets, relaxed apo-h and relaxed holo-h, were produced in 500 ns cMD simulations of apo-h and holo-h forms so that during the first 100 ns of cMD a H-bond Q78···G62 was prevented; then, both forms were simulated with the fully lifted restriction.

By comparing the relaxed models with the models stabilised by the H-bond Q78···G62, we observed that the H-bond alternation (its presence or absence) is connected to L-loop folding (Figure 5). Despite the cap secondary structures conservation in both models (H1-L helix in apo-h and a random coil with two turns in holo-h), (i) an H2-L helix formed in apo-h over GaMD simulation, was preserved in the relaxed apo-h model only during the first 100 ns (with H-bonds prevented), and, later (ii) (with the fully lifted restriction on the L-loop ends), was transformed into short-lived 3_10_-helix, which was further partially denatured in a coil; (iii) two adjacent 3_10_-helices (H3-L) were converted into a unique helix folded as an α-helix in the relaxed apo-h and a 3_10_-helix in the relaxed holo-h; (iv) H4-L helix was also unstable in two forms, reversibly transiting from α-helix to a coil. As a result, the number of folded structures (α- and 3_10_-helices) in the L-loop of each relaxed model was not identical to the forms stabilised by the H-bond Q78···G62 (Figure 4c,d and Figure 5b,c). Each relaxed form shows a clear evolution of its folding, which increased in apo-h and reduced in holo-h, demonstrating a tendency to be comparable in both forms.

Curiously, the expected restoring of the H-bond Q78···G62 upon the fully lifted restriction was not observed during the simulation of any hVKORC1 forms studied.

In addition to the evident change in L-loop folding, the relaxed forms demonstrate significant conformational mobility, viewed by the displacement of their helices with respect to the TMD, and the decrease of the distance between centroids defined on the cap and hinge of the L-loop was especially noticeable in the relaxed apo-h model (Figure 5e). Conformations of the relaxed forms showed that a temporary restriction of H-bond Q78···G62 promotes the displacement of two L-loop regions, hinge to cap, and that this effect is more pronounced in relaxed apo-h form. The proximal position of the cap and hinge was observed early in the predominant L-loop conformation of the de novo model [14].

Together with the significant structural and conformational reorganisation, increasing dynamical coupling between the TMD and L-loop was observed only in apo-h form with the relaxed H-bond Q78···G62, as showed by the cross-correlation maps (Figure 5g). The strong positive correlations inside TMD, early attributed to the collective drift of TM helices [14], were observed in all models.

Similar to the increased RMSF values, the first two PCA modes denote essential collective motions of the L-loop that are greater in the relaxed forms. The two first modes characterisee most of the L-loop motion displaying, large-amplitude collective movements of helices and adjacent coiled linkers (Figure 5h). The amplitude and direction of L-loop motion in relaxed models of each form, especially apo-h, are increased compared to the forms stabilised by the H-bond Q78···G62, suggesting a larger conformational space for the relaxed L-loop.

### 2.4. Is hVKORC1 a Modular Protein?

hVKORC1’s main functions—reversible activation/deactivation over the thiol–disulfide exchange with its redox protein and recurrent reduction of vitamin K—involve L-loop and TMD, respectively. Our previous studies [14,15], and results reported above in this work, have shown that (i) transient secondary structures and high conformational variability qualify the L-loop as an intrinsically disordered region, (ii) the TMD structure is strictly ordered, and apparently do not dependent on L-loop disorder, and (iii) there are no long-lived H-bond interactions between the TMD and L-loop. These hVKORC1 properties observed in all studied models, except for limited conformational flexibility of the L-loop in the crystallographic structures, suggest that hVKORC1 is a modular protein and that the L-loop and TMD are two of its structural and functional subdomains.

The ability of modular protein domains to independently fold and bind proteins has allowed a significant number of protein–protein interaction studies in silico, in vivo, and in vitro performed on isolated modules used as more accessible items [19].

To investigate this domain as a suitable promoter in the activation/deactivation process of hVKORC1 by its redox protein, we studied L-loop as an isolated polypeptide cleaved from TMD. The slightly extended L-loop (R33-N80) was cleaved from (i) the homology models of hVKORC1, holo-h and apo-h, generated from two crystallographic forms of the oxidised state, (ii) their derivatives with the relaxed H-bond Q78···G62, and (iii) the de novo model of the same enzymatic state. The initial models of L-loop differ considerably as shown by the RMSD values, which are smaller between the apo-h and holo-h forms (of 4.0 Å) than between each form and de novo model (of 8.4 Å) (Figure 6a).

First, the cleaved polypeptides were studied by the conventional (cMD) and accelerated (GaMD) simulations with the soft restrains (elastic retains) to maintain the flexible ends of each polypeptide at the distance observed in VKORC1. Restriction on the L-loop ends distance mimics the steric conditions imposed by the enzyme transmembrane domain. Although using constraints upon the MD simulation with constant pressure is a questionable subject [20], the applied restriction on L-loop ends allowed the distance preservation between R33 and N80 residues in the range of 15–20 Å, comparable with the distance of 15 Å in the empirical structures [16] and of 16 Å in the de novo model [14].

The extended (5-μs) cMD simulation of cleaved L-loop has generated more similar conformations between apo-h, especially in its relaxed version, and the de novo model than between two forms derived from crystallographic structures (Figure 6). Furthermore, the L-loop cleaved from apo-h and relaxed apo-h forms and the de novo model show (i) close RMSD profiles and values and (ii) a very similar portion of ordered (helical) structures (40, 38, and 38% in apo-h, relaxed apo-h, and de novo, respectively), which is twice higher than in holo-h form (21%). Despite the equal number of helices (four helices), the principal difference between apo-h and the de novo model is that the structural organisation of the cap showed a large H1-L helix in apo-h and relaxed apo-h and a poorly folded or unfolded segment in the de novo model; the other helices are rather different in type (α- and 3_10_-helices) and length. Nevertheless, in all studied cleaved L-loop models, all helices are transient, reversibly conversed between α- and 3_10_-helix (α-helix ↔ 3_10_-helix) and between 3_10_-helix and turn or bend (3_10_-helix ↔ turn/bend). These recurring transitions of the L-loop secondary structures over the trajectory promote helices with great length variation. Proportions of folded structures (α- and 3_10_-helices) in the cleaved L-loop are identical to the L-loop in the corresponding model of hVKORC1 (Figure 4d, Figure 5c and Figure 6).

Remarkably, even without any condition, the H-bond Q78···G62 stabilising the L-loop hinge has definitely broken at 4.3 µs (apo-h) and 2.2 µs (holo-h) of the cMD simulation of the cleaved L-loop. Curiously, the H-bond rupture was spontaneous and apparently unrelated to the variation in distance between the N- and C-ends of the L-loop. The H-bond disruption is accompanied by several structural effects displayed as (i) a change in the folding of residues R61 and N80, (ii) a complete unfolding of the H3-L double helix, (iii) a tendency of H2-L to fold as a 3_10_-helix, and (iv) the stabilisation of the long H1-L helix.

These transformations in the cleaved L-loop lead to an overall change in its shape, which tends to be more compact in both shapes and approximates the L-loop prevalent shape in the de novo model (Figure 6g). Similar results were observed over a GaMD simulation of the same length (data not shown).

Secondly, we investigated whether the L-loop structure and conformations depend on steric conditions imposed on N- and C-ends, either naturally occurring in hVKORC1 or mimicked by soft constraints on cleaved L-loop terminal residues. The cleaved L-loop (apo-h, holo-h, relaxed apo-h, and de novo) was studied by 0.5 μs cMD simulations without any constraints.

RMSD and RMSF values of each fully liberated polypeptide vary within ranges observed for the L-loop either fused to TMD or simulated as a cleaved polypeptide under constraints (Figure 7). Similar to the restrained L-loop, the released L-loop folding (secondary structures) was transient in each form, and folding order was comparable with the end-restrained L-loop. In particular, the total of folded structures in the cleaved L-loop simulated with constrains or not was equivalent in the apo-h (40/40%), holo-h (22/21%), and relaxed apo-h (38/37%) but decreased in the de novo model (38/25%). Curiously, some structural effects, in particular the alternative structural content of residues R61 and N80 and full unfolding of the double H3-L helix, were observed in the apo-h L-loop simulated under both conditions on the terminal residues, either as the end-restrained sample or as the released polypeptide.

The distance between the L-loop ends in apo-h form was 21 Å during the first 250 ns of simulations, further gradually decreased to 13 Å, and was maintained until the end of the simulation. Similar variations of this distance were observed in apo-h form and the de novo model simulated without hindrance. It should be noted that, in many conformations of apo-h and de novo, the N-and C-ends distance values in the unrestrained L-loop agreed well with the L-loop fused to TMD. In holo-h form, this distance varied reversely in a very large range, from 5 to 35 Å. The shape of the unconstrained L-loop was similar to related constrained forms (Figure 6 and Figure 7).

Comparing each form of cleaved L-loop, simulated either under constrained N- and C-ends or fully released species, with the L-loop fused to hVKORC1, we affirm that in all cases, the L-loop is an intrinsically disordered region having (i) a helical fold with a quasi-equal content of folded structures, (ii) a similar sequence position of transient helices (H1-H4), and (iii) a comparable overall 3D shape. Consequently, the L-loop’s structural properties—folding, conformations, and degree of the intrinsic disorder—are independent from its physicochemical context as the entity. We postulate that L-loop folding is principally sequence dependent and is not controlled by the hVKORC1 transmembrane domain. At the same time, TMD restricts the L-loop ends distance, probably optimising L-loop geometry.

Consequently, this analysis clearly confirms that the hVKORC1 modular structure is composed of the quasi-rigid and stable transmembrane domain and intrinsically disordered L-loop. The TMD stability is maintained by an extended network of non-covalent interactions (Appendix A) organising four TM helices in a structural motif described as the alpha-helical coiled coil [21,22]. This motif is conserved in hVKORC1 inserted into the membrane or placed in an aqueous solution. Indeed, during hVKORC1 MD simulations in different environments, the TMD helices do not vary in structure and are held together by similar or strictly identical non-covalent interactions supplying the TMD a tight packing in each model.

### 2.5. How Are L-Loop Conformations Similar?

Comparing time-related folding, we observe that transient events in different segments of the L-loop sequence may be mutually related or not. Thus, the ordering of one helix favours an unfolding of the other, as seen in the de novo model or holo-h form of the L-loop simulated with restrained N- and C-ends (Figure 6e). On the other side, holo-h form conformations display H2-L unfolding (from ~1 µs) followed by a H3-L transition to a coil (from 2 µs), synchronous with H4-L unfolding. The significant decrease in holo-h L-loop folding is partially compensated by the H1-L increase in length and transient H3-L folding as the α-helix. Nevertheless, the helical content of this form is obviously lower than in other studied L-loop entities. The folding of distinct segments in apo-h and relaxed apo-h forms shows different folding–unfolding effects relationships—interrelated, disconnected, and their combination. Therefore, interrelations between folding–unfolding effects in a disordered L-loop are apparently more sophisticated than their description at the secondary structures level and probably also depend on global and local flexibility.

To study two structural processes in the L-loop, folding and flexibility, and their interrelations, we tried to regroup similar conformations of the cleaved L-loop simulated with constrained N- and C-ends, as they represent the richest data (5 µs trajectories). A search for the similar MD conformations of the L-loop from different forms of VKOR was performed on the concatenated trajectories (apo-h, holo-h, relaxed apo-h, and de novo). Primarily, we found that L-loop conformations with minimal RMSD values (4–5 Å) show shape similarity, while their secondary structures are highly different (Figure 8). Secondly, the secondary structures-based clustering [23] regrouped MD conformations (0.65 was chosen as the most appropriate value from the tested cut-off: 0.5, 0.6, 0.65, 0.7, and 0.8) into seven clusters (C1-C7). Clusters C5 and C6 contain conformations derived from only one L-loop form (respectively, apo-h and apo-h relaxed); clusters C1–C4 and C7 comprise conformations from several L-loop forms (Figure 8b). Only clusters C2 and C4 are constituted by conformations generated over all four trajectories.

We observed that (i) the secondary structures of conformations regrouped in a cluster are only partially similar, usually showing a resemblance for the one or two L-loop segments, while others differ significantly; (ii) conformations of each cluster C2, C4, or C7 are either similar in shape (C4 and C7) or largely distant (C2); and (iii) independent from shape similarity, conformations within each cluster are highly different (RMSD of 6.5–10.0 Å) (Figure 8c,d).

The results produced by RMSD and secondary structures-based clustering showed that both approaches are not suitable for the analysis of intrinsically disordered L-loops. We searched more intuitively/manually for L-loop conformations with both the minimum RMSD and close structural similarities, which resulted in finding more comparable conformations than those obtained using automatic clustering based on a single criterion.

Thirdly, concatenated data combining cMD trajectories of the L-loop cleaved from apo-h, holo-h, relaxed apo-h forms, and the de novo model were analysed with principal component analysis (PCA). The projection of the generated conformations on the principal components PC1-PC3 shows partially overlapped subspaces (Figure 8d). These overlapped areas relay, successively, all subspaces and form compact generic ensembles of L-loop conformations. Evidently, the generic ensemble combined from subspaces does not represent a full conformational space of the disordered L-loop but reflects more exhaustively its conformational properties than a unique subspace.

The conformational space of the cleaved L-loop simulated with unrestrained N- and C-ends is less compact and significantly less explored (by factor 10). Nevertheless, similar to a constrained L-loop, each subspace overlaps with another, and the subspace formed by relaxed apo-h conformations showed an overlapping with apo-h and holo-h forms.

Clearly, the cMD simulation of each cleaved L-loop generates only a limited part of the intrinsically disordered L-loop’s overall conformational space. We suggested that the combined representation of these limited portions using the free energy landscape model will allow a better characterisation of the generated set, if it is not yet complete. We are aware that such a representation does not provide the free energy quantitative characteristics, but it can be useful for the comparison of different datasets. The relative Gibbs free energy, ΔG, defined on chosen coordinates called reaction coordinates, describes a protein conformations between two or more states, measured as the probability of finding the system in those states [24].

For the relative free energy (ΔG) evaluation and reconstruction of L-loop conformational ensemble landscape, primary measures—radius of gyration $\left(Rg\right)$ and distance (RMSD)—were used as reaction coordinates. The free energy landscape (FEL) as a function of RMSD and radius of gyration Rg ($FEL_{RMSD}^{R_{g}})$ was characterised for the concatenated data of cMD trajectories of L-loop cleaved from each form of hVKORC1 (holo-h, apo-h, relaxed apo-h, and de novo model). The $FEL_{RMSD}^{R_{g}}\;$ determined on normalised conformations shows two very closely positioned, narrow, deepest potential wells, W1 and W2, complemented by W3, an adjacent well satellite (Figure 9). All these very proximal wells, separated by the InfleCS* method [25], are composed of compact globule-like conformations generated from the relaxed apo-h form and de novo model (W1), relaxed apo-h and apo-h forms (W2), and a mixture of apo-h, relaxed apo-h and de novo (W3). Conformations from these wells are similar principally by size (Rg values ~10–10.5 Å) and shape (closed form of L-loop) and apparently are enabled to reversible transition as viewed by a low ΔG barrier on $FEL_{RMSD}^{R_{g}}$.

Shallow flattened wells W4 and W5 include L-loop conformations derived from all analysed samples. These wells L-loop conformations differ mainly in shape and size, with a compact, globular-like L-loop (closed conformation) in W4 and an elongated L-loop (open conformation) in W5. Despite the conformations’ similar shapes in W4 and W1-W3, higher RMSD values in W4 are a discriminating factor leading to the separation of W4.

Both analytical methods, PCA and the reconstruction of the relative free energy landscape using the primary measures—radius of gyration $\left(Rg\right)$ and distance (RMSD)—as two reaction coordinates, showed that the sampling of the L-loop by multiple independent MD simulations from largely different initial molecular conformation originated either from empirically determined structures or theoretically predicted models, converging to similar L-loop conformations. The study of cleaved L-loops accesses this convergence more rapidly than for the entire protein, even sampled by GaMD simulation.

### 2.6. Protein–Protein Docking: PDI and VKORC1 as the Partners

To evaluate hVKORC1 conformations as putative targets for Protein Disulfide Isomerase (PDI) suggested as a redox partner [15], we used High Ambiguity Driven protein–protein DOCKing (HADDOCK) [26]. Unlike other protein–protein docking approaches, based on the combination of energetics and shape complementarity, HADDOCK uses biophysical interactions data—in our case, a short distance between sulfur atoms from the cysteine residues of two interacting protein, PDI and hVKORC1, to drive the docking process.

In the docking analysis, PDI *per se* is an invariable component taken from [15], while hVKORC1 is a variable item that can be any randomly chosen conformation generated by cMD simulation of the theoretical model (de novo) or crystallographic structure. We suggested that using of different target conformations will help discriminate an authentic conformation specific to its ligand. This study was carried out with aim of answering the following essential question: what conformation of hVKORC1 is an optimal target for PDI?

Prior to docking studies, we performed a bench test to investigate if docking with HADDOCK can reproduce the predicted de novo PDI–hVKORC1 complex. Docking trials were performed using the published de novo structural model (Model 1) of the PDI–hVKORC1 complex [15] as a benchmark set. The theoretical model (de novo 3D model) application as a reference is imputable to the absence of empirical structural data for the PDI–hVKORC1 complex. The PDI–hVKORC1 complex (Model 1) was separated into unbound proteins and docked with HADDOCK and Ambiguous Interaction Restraints (AIRs) [26] using a pair of cysteine residues thiol groups, C37 from PDI and C43 from hVKORC1, as active centres to drive the docking process. For objectivity, each protein was considered as a target and as a ligand.

The docking of PDI as a ligand into hVKORC1 as a target (scenario i) showed two clusters, C6 and C9 (numbered by HADDOCK), formed with models of PDI–hVKORC1 complexes similar to a benchmark structure (Appendix A). The docking of hVKORC1 as a ligand into PDI as a target (scenario ii) did not lead to a benchmark solution. In both scenarios for a ligand–target pair, (i) and (iii), both docked proteins structures and conformations were well conserved, with maximal RMSD values of 0.4 (PDI) and 1.0 Å (hVKORC1).

Curiously, two alternative scenarios in the docking of the ligand–target pair gave rise to largely different solutions. Focusing on solutions produced for case (i), we noted that PDI is located above the L-loop and occupies an approximately similar spatial position for HADDOCK solutions and the benchmark, changing only its orientation resulting from the rotation of PDI around the central hVKORC1 axis. The only exception was cluster C10, where PDI was located on one side of the L-loop in a position perpendicular to the hVKORC1 central axis. In case (ii), the position of PDI on one side of the L-loop was observed in most docking solutions (6 clusters out of 10). Since such solutions are not compatible with the membrane position, they have not been considered. A HADDOCK solution was classified as interpretable if the PDI position matched the Trx domain position in the hVKORC1 bacterial homolog, and human VKORC1 uses the same electron transfer pathway as its bacterial homologues [8]. Docking trials showed that HADDOCK reproduces the benchmark model (Figure 10).

Comparing the obtained results for two docking scenarios, we found that scenario (i) is the right choice leading to the benchmark solution. Therefore, this scenario was further used in docking examinations.

Two strongly different hVKORC1 conformations with the L-loop compact (closed, most probable) and elongated (open, least probable) shapes were randomly chosen from data generated by cMD simulations of the de novo model of hVRORC1 [15] and used as targets for PDI docking. The docking results show that both conformations of hVKORC1, with elongated (open) and compact (closed) L-loops, produced solutions where PDI is positioned in the same space volume as in the benchmark complex; however, PDI orientation with respect to the target was highly divergent.

The HADDOCK quantitative metrics of binding modes—number of clusters, score, and population—do not allow comparisons between solutions for open and closed conformations, even if a simple superimposition of found solutions showed no benchmark solution was observed. Surprisingly, HADDOCK solutions for both hVKORC1 forms showed close similarity to the alternative PDI–hVKORC1 complex Model 2 proposed in [15] as a possible solution (Figure 10). Moreover, the number of such interpretable solutions is greater for hVKORC1 with an L-loop in closed conformation.

PDI docking onto hVKOR models apo-h and holo-h derived directly from crystallographic structures [16] did not produce the expected benchmark solution, nor solutions corresponding to the alternate model of the PDI–hVKORC1 complex. Moreover, the majority of HADDOCK solutions have low compatibility with the bacterial homologue of hVKORC1, the unique empirical structure in which the VKOR and Trx-like domain are covalently bound [8].

Finally, to test if cleaved L-loop is a valid target for PDI docking, PDI was docked onto open and closed L-loops cleaved from the respective de novo models. These docking experiments found, again, HADDOCK solutions corresponding to Model 2 and showed that (i) solutions are very similar to PDI docking onto the full-length hVKORC1, and (ii) a closed, cleaved L-loop is the best target of PDI (Figure 11). Similarly, PDI docking into the L-loop cleaved from hVKOR models apo-h and holo-h derived directly from the crystallographic structures [16] did not produce any reasonable solution.

## 3. Discussions

As vitamin K is involved in various crucial biological processes [27,28], in particular in blood coagulation, an explicit understanding of each step leading to its transformation by hVKORC1 is required to control such processes in the context of its deregulation activity leading to severe diseases [29,30]. At a molecular level, the formation of a hVKORC1-redox protein molecular complex is a fundamental process leading to thiol-based redox switches, occurring primarily at an inter-protein level between cysteine residues of redox proteins and the L-loop and, further, inside hVKORC1, between cysteine residues of the L-loop and the highly conserved CXXC motif, to transform vitamin K epoxide to its reduced form, hydroquinone [6].

The PDI–hVKORC1 molecular complex, reported as a first precursor to probe the thiol–disulfide exchange reactions between hVKORC1 and PDI [15], was modelled with PDI targeting the native hVKORC1 de novo model, a transmembrane four-helix bundle crowned by an intrinsically disordered luminal loop, protruding in the endoplasmic reticulum lumen [14]. Published X-ray VKOR structures with terminals restrained by green fluorescent protein [16] mostly confirm the de novo model correctness. First, they delivered a solid empirical affirmation of the VKOR four-helix TM domain, initially predicted by de novo modelling. Second, the conformational transition reported in [16] is a strong argument of L-loop flexibility observed in the de novo model. Nevertheless, a careful analysis of MD simulation data obtained for the human VKORC1 homology models built from crystallographic structures reported two forms of the enzyme’s oxidised state, demonstrated the L-loop curious fixedness, and raised questions of the L-loop role in VKOR activation. L-loop rigidity conflicts with its primary functions—recognition and binding of redox proteins—for which the L-loop must be easily adaptable and highly flexible. Moreover, the L-loop’s expected conformational transformation, explained by the authors in terms of an open-to-closed transition [16], was not observed under the MD simulations of two forms in identical conditions (unbound protein in water solution).

Our search for sterical and physical conditions required for L-loop transition identified the intra-L-loop H-bond, Q78···G62, as a main factor leading to L-loop rigidity. Even a short-lived constraint preventing this H-bond formation increased L-loop flexibility in both hVKORC1 forms delivered from crystallography. Moreover, extended cMD simulations of an isolated L-loop, cleaved from crystallographic forms, showed a per se rupture of this H-bond. Apparently, the Q78···G62 H-bond stabilises certain L-loop conformations under particular circumstances (e.g., protein crystallisation conditions), but, generally, the hVKORC1 L-loop poses great flexibility. This L-loop quality is mandatory and determined by its functional role to easily adapt its conformation in response to an external stimulus (redox protein) or biochemical (e.g., thiol–disulfide reaction) factors.

It has been established that increased intrinsic plasticity represents an important prerequisite for effective molecular recognition [31,32,33,34], and long-range conformational changes mediates enzymatic reactions [35].

As hVKORC1 is an enzyme-using thiol–disulfide reaction for its activation by a redox protein followed by protons transfer to the active site for vitamin K processing, the hVKORC1 intrinsically disordered L-loop is an excellent and optimal platform to ensure this complicated multi-step reaction. This biochemical process requires transformations between the functional groups –SS–, –SH, and –S• of two proteins, hVKORC1 and its redox protein, during the transfer of two protons/electrons. The kinetics and mechanisms of thiol–disulfide substitution and redox reactions having a pivotal role in biology are well-described for different small molecular and enzymatic systems [36,37]. In particular, a computational study of thiol–disulfide exchange reactions reactivity in thioredoxins and in other proteins concluded that these reactions are critically fine-tuned by the active site atomistic details [38]. A prime example is the hydrophobic pocket around the thioredoxin family CXXC motif, the geometry, dynamics, and electrostatic environment of which decide on the redox potential and kinetics [39]. Given the multitude of possible thiol–disulfide exchange reactions, an important but *hitherto* unresolved question is how specificity is achieved. Nevertheless, the highly dynamic disordered nature of regions playing a pivotal role in such reactions was systematically reported [40].

In this context, we address the central question—is hVKORC1 L-loop a properly folded, two-state domain providing a reversible open-to-closed conformational transition (on-off switch), as postulated in [16], or is it an inherently disordered region the possessing great plasticity/adaptability required to perform various steps during the activation process?

All data generated by the MD simulation of two different crystallographic forms, their models derived from these structures and the de novo model, confirmed that the L-loop intrinsic disorder consists of two reversible processes—transient folding and high conformational flexibility, leading to the L-loop’s enormous conformational diversity, its conformations ranging from a closed, compact globule-like shape (the most prevalent), to a rare open, elongated boat-shape, through a limitless number of intermediaries. All structurally organised L-loop segments are involved in the reversible folding–unfolding process (structural transitions), and the L-loop exhibits great conformational flexibility supplied by linear and rotational displacements and their combination in either folded or coiled segments.

The L-loop, as a highly disordered region, possesses large conformational plasticity, supplying a great capacity for the multiple structural and conformational arrangements required for different steps of the thiol–disulfide exchange reaction leading to hVKORC1 activation. Such an interpretation of the hVKORC1 structure does not contradict an increased propensity of the L-loop to be disordered. The L-loop sequence contains a large part of polar and charged residues (54%), while the hydrophobic residues portion is reduced to 44%, a typical composition of ID proteins [34,41]. It was shown that L-loop flexibility depends on the oxidation/reduction state of hVKORC1, and the L-loop in its fully oxidised (inactive state) is considerably less flexible and more folded compared to the reduced states [14]. Nevertheless, an ‘oxidised L-loop’ demonstrates a remarkable structural and conformational plasticity, evidenced by the variation of its helices in the length and spatial positions, giving rise to myriad L-loop conformations. As L-loop transient folding (at the secondary structures level) was observed in the quasi-rigid (apo-h and holo-h) and flexible (relaxed apo-h and de novo) species, this process is possibly disassociated from L-loop conformational flexibility.

As two kinds of L-loop disorders—transient folding and conformational flexibility—do not appear to affect hVKORC1 TMD structure at a given (oxidised) state, we examined if VKORC1 can be considered a modular protein composed of the transmembrane domain and L-loop as two sub-domains of the enzyme.

According to the obtained results, hVKORC1 is composed of the stable transmembrane domain and intrinsically disordered L-loop. The TMD stability is due to multiple inter-helices, non-covalent interactions—H-bonds and van-der-Waals contacts—which maintain four transmembrane helices together, forming a coiled coil [21]. This structural motif is perfectly preserved in hVKORC1 inserted into a membrane or placed in an aqueous solution, as was shown by classical or accelerated MD simulations. The location of hVKORC1 on the endoplasmic reticulum membrane is crucial for its proper folding [42], and once it was already folded, we hypothesised that the membrane does not affect the hVKORC1 structure maintained by preformed interactions within the TMD. Moreover, the TM helices position is well maintained on a membrane, showing only collective drift [14].

The absence of stable, non-covalent contacts between the well-ordered, highly stable TMD and fully disordered L-loop, and their limited dynamical coupling, is a good argument to conceptualise hVKORC1 as a modular protein. Moreover, a cleaved L-loop and L-loop fused to the TMD showed very comparable structural and conformational properties, as was confirmed by different simulation methods (cMD and GaMD) and conditions of simulation (in membrane or water solution only). In all studied cases, the fully oxidised hVKORC1 L-loop adopts a compact, globular-like shape (closed conformation), the most prevalent and apparently the most energetically favourable conformation of the L-loop in solution. Therefore, the role of the L-loop hinge, which, according to [16], is partially embedded in the membrane and acts as an anchor, requires detailed study.

One approach for studying the assembly of multidomain proteins and their folding is to use the protein modular domain, which preserves binding capabilities even when the domain is removed from the full-length protein context [19]. Dynamic modularity displays clear sub-domain architectures that give protein modules enhanced flexibility and might influence their ability to respond to the redox protein selection [43,44]. Modular protein domain’s ability to independently fold and bind both in vivo and in vitro has been taken advantage of by a significant portion of proteomics studies to assess protein–protein interactions required for a diverse set of biological processes. The L-loop application (empirically and numerically) as a cleaved polypeptide represents a promising strategy for hVKORC1 thiol–disulfide exchange reactions study and the modulation of protein function by the controlled interference of underlying molecular interactions. Such use is fully justified, as we proved that the hVKORC1 L-loop is a modular domain that preserves perfectly structural and dynamic properties upon its fused and cleaved status.

These considerations are valid for an enzyme given state performing its specific function(s). The hVKORC1 principal function in its fully oxidised state consists of the recognition and binding of its redox protein, leading to inter-protein thiol–disulfide exchange reactions. Most likely, this function is performed exclusively by the L-loop. The next step, intra-hVKORC1 thiol–disulfide exchange reactions that result in the activation of the CXXC motif-forming hVKORC1 active site, requires tight cooperation between the L-loop and transmembrane domain on all levels—structural, dynamical (structural/conformational transitions), physical (electron transfer), and biochemical (bond cleavage/formation). Vitamin K transformation in the active site can be realised without L-loop participation, while the last step of hVKORC1 catalytic cycle—restitution of the initial oxidase state—requires, again, L-loop contribution as a principal actor. Consequently, the hVKORC1 structure can be conceptualised as a context-aware, self-organising system, providing global feedback to modulate and coordinate vitamin K transformation.

The L-loop rich conformational space promoted by two different processes—transient folding and high flexibility—creates a serious problem in grouping L-loop conformations into clusters containing similar conformations. RMSD-based clustering methods, typically used to define prevalent protein structure, depend on a proper cut-off between cluster groups to be effective [45]. With this sensitivity to input parameters and low separation dimensionality, RMSD-based methods are not the best choice for the analysis of disordered proteins that sample a large conformational space. Moreover, RMSD similarity calculated in Cartesian coordinate space is sensitive to how the structures are aligned beforehand and ignores real deviations within the backbone. Similarly, the secondary structure-based hierarchical clustering local alignment algorithm [23,46] is not the best solution for disordered proteins. The application of these two different approaches to L-loop clustering is not crowned with any success. The difficulty is to regroup disordered L-loop transient structures in a way that is reductive enough to provide required simplification while being flexible enough to accommodate a wide range of irregular structural configurations [47].

To avoid or reduce the problem, we decided to represent L-loop conformational space by the reconstruction of its free energy landscape using as reaction coordinates the primarily descriptors—radius of gyration and RMSD. As the molecular dynamics simulation of the L-loop cleaved from distinct forms of hVKORC1 produced conformational subspaces that are partially overlapped, we suggested that their combining would be an appropriate approach to represent the generic L-loop conformational space, even if it is not yet complete. The free energy landscape, modelled on cumulative conformational space, specifies qualitatively the shape, folding, and dynamics of the disordered L-loop and allows for comparison between its different configurations. The L-loop free energy landscape showed some local minima populated by ensembles of quasi-isoenergetic L-loop conformations typically seen in disordered proteins [48].

As we analysed hVKORC1 as a target of its redox protein, in particularly focusing on its ability to form protein–protein interactions mandatory for thiol-based redox switches, we concentrated on the central question—what conformation of the VKORC1 oxidised state is an authentic target of redox protein PDI?

The mechanisms of folding coupled to binding are poorly understood, but it has been hypothesised on theoretical grounds that binding kinetics may be enhanced by a ‘fly-casting’ effect, where the disordered protein binds weakly and non-specifically to its target and folds when approaching the cognate binding site [49]. Consequently, ID protein capable of adapting to binding surfaces through coupled folding and binding [50] and the same binding region may have the capacity to bind several different partners with very similar affinities [51].

Similar to our previous paper, we have taken PDI as the hVKOR-interacting redox partner, although this question remains open to discussion and still awaits empirical identification [8,52,53]. Docking of PDI (ligand) onto hVKORC1 (target), performed with HADDOCK preliminary tested on the PDI–hVKORC1 complex [15] as a benchmark, clearly indicated that the most interpretable solutions were found for the L-loop closed form used as a target only. Note that a HADDOCK solution was classified as interpretable if (i) the PDI position matched the hVKORC1 bacterial homologue thioredoxin-like domain position; (ii) human VKORC1 was shown to use the same electron transfer pathway as its bacterial homologues [8]. Surprisingly, the interpretable HADDOCK solutions did not correspond to the benchmark complex, but to the other alternative model (Model 2) of the PDI–hVKORC1 complex, reported in [15]. Given two probable solutions for the PDI–VKORC1 complex (Model 1 and Model 2), this would seem to be an area with substantial potential for further development.

As modularity provides biological systems with a convenient way to present binding sites on stable protein scaffolds, in the right position for the function, and also allows for regulation by module rearrangement [54], we investigated whether a cleaved L-loop separated from the TMD will retain its fused context scaffolding properties. PDI docking onto the L-loop as a target produced solutions similar to those obtained by PDI docking onto hVKORC1. In particular, we found that (i) only L-loop closed conformation allows for the recognise and binding of PDI, and (ii) PDI docking onto L-loop produced again the most interpretable solutions corresponding to Model 2. Considering cleaved and fused L-loop similar properties in folding and conformational plasticity, and also the capacity to recognise PDI, we suggested that a cleaved L-loop is a convenient entity in studies of hVKORC1 recognition/activation by its redox protein.

Also, the application of hVKORC1 (the membrane protein) in aqueous solution, as shown here, is likely to prove to be very useful in practice in either in silico studies or in vitro experiments. Although, for today, there are no empirical data available for a complex of hVKORC1 with its redox protein, our results can be useful to engender working hypotheses for such studies.

## 4. Materials and Methods

### 4.1. Available X-ray Data and Homology Models

***X-ray data***: The original atomic coordinates of two crystallographic forms of VKOR in the inactive state, the holo-c form, obtained by the co-crystallisation of human protein with an inhibitor (warfarin) and glycerol monooleate (PDB ID: 6wv3), and the apo-c form that was received for the ligand-free VKOR-like protein from *Takifugu rubripes* (PDB ID: 6wvi) were retrieved from the Protein Data Bank (PDB) [17]. These data were used (i) for the comparative analysis of protein structures (holo-c *versus* apo-c), (ii) as a template for homology modelling of the human protein (apo-h), (iii) for the optimisation of the holo-c form (repair the missing residues) to obtain the full-length protein (holo-h), and (iv) as the starting conformation for the MD simulation of the holo-c.

***Homology models***: Three-dimensional homology models of the full-length human VKOR (1–163 aas), apo-h, and holo-h, were generated. The apo-h model was obtained with Modeller [55] from the human sequence Q9BQB6 (https://www.uniprot.org/uniprot/, accessed on 10 February 2022) and the empirical structure 6wvi *(VKOR-like protein from Takifugu rubripes)*, used as a template. The similarity/identity of two sequences is 69/46% for the total protein and 77/65% for the L-loop. The holo-h model was obtained from the holo-c form by adding of the missing residues/atoms.

***Relaxed homology models***: The relaxed hVKOR models, relaxed apo-h. and relaxed holo-h, in which the H-bonds G62···Q78 and G60···N80 were removed using the translations of the neighbouring residues R58-W59 (x, y + 4, z − 2), G60-F63 (x, y + 3, z − 2), and Q78-S79 (x, y − 2, z) were prepared. The coordinates of the TM3 helix (W101-V127) were also translated using (x, y + 2, z) to avoid steric clashes.

***Models of the isolated L-loop***: The coordinates of the L-loop, slightly extended in the sequence at its C-end (R33–N80), were extracted from the holo-h, apo-h, and de novo models and used as the starting conformations of the cleaved isolated L-loop (holo-h L-loop, apo-h L-loop, and de novo L-loop).

The stereochemical quality of 3D model was assessed by Procheck [56], which revealed that more than 96% of nonglycine/nonproline residues have dihedral angles in the most favoured and permitted regions of the Ramachandran plot, as is expected for good models.

### 4.2. Molecular Dynamics Simulation

#### 4.2.1. Set-Up of the Systems

In each system, the structure of the holo-c form, the homology models apo-h and holo-h, their relaxed models, relaxed apo-h and relaxed holo-h, and models of the cleaved L-loop were prepared with the LEAP module of Assisted Model Building with Energy Refinement (AMBERTools 20) (http://ambermd.org/AmberTools.php, accessed on 10 February 2022) [57] using the ff14SB all-atom force field parameter set [58] and TIP3P water models: (i) hydrogen atoms were added; (ii) covalent bond orders were assigned; (iii) protonation states of amino-acids were assigned based on their solution for pK values at a neutral pH, and the histidine residues were protonated on their ε-nitrogen atoms; (vi) counter-ions, Cl^-^, were added to neutralise the charge of each protein; (v) each protein was placed in an octahedron water box. Each final system contained 2595 atoms of VKOR and 45987/57222 atoms of water for the apo/holo forms, respectively, and 741 atoms of L-loop and 19302/18876 atoms of water for the cleaved L-loop.

#### 4.2.2. Minimisation and Equilibration of the Systems

Each system was minimised and equilibrated using the Sander module of AmberTools20 using the steepest descent and conjugate gradient algorithms through the 30,000 minimisation steps as follows: (i) 10,000 minimisation steps where water molecules were fixed, (ii) 10,000 minimisation steps where the protein backbone was fixed to allow protein sidechains to relax, and (iii) 10,000 minimisation steps without any constraint on the system. A 100 ps thermalisation step was performed, where the temperature (atoms velocity) was gradually increased from 0 to 310 K using the Berendsen thermostat with imposed periodic boundaries conditions and isotropic position scaling [59]. Then, a 100 ps equilibration with constant volume (NVT) and a 100 ps equilibration with constant pressure (1 bar) (NPT) were performed. For these two steps, temperature regulation was performed with Langevin dynamics with a friction coefficient γ = 1. Finally, a 100 ps molecular dynamics was completed at 310 K (Langevin dynamics), constant volume, and constant pressure (hybrid Monte Carlo barostat [60]. All equilibration steps were carried out with an integration step of 2 fs. Non-bonded interactions were calculated with the Particle-Mesh Ewald summation (PME) with a cut-off of 10 Å, and bonds involving hydrogen atoms were constrained with the SHAKE algorithm [61].

#### 4.2.3. Production of the Conventional Molecular Dynamics Trajectories

The conventional Molecular Dynamics (cMD) trajectories of the holo-c structure (0.2 µs), the homology models apo-h and holo-h (0.5 µs), their relaxed models (relaxed holo-h and relaxed apo-h) (0.5 µs), and models of the cleaved L-loop with restrains (5 µs) and fully relaxed (0.5 µs) were generated using the AMBER ff14SB force field with the PMEMD module of AMBER 16 and AMBER 18 (GPU-accelerated versions) [57] running on a local hybrid server (Ubuntu, LTS 14.04, 252 GB RAM, 2x CPU Intel Xeon E5-2680 and Nvidia GTX 780ti) and the supercomputer JEAN ZAY at IDRIS (http://www.idris.fr/jean-zay/, accessed on 10 February 2022).

The initial velocities were reassigned according to the Maxwell–Boltzmann distribution. A time step of 2 fs was used to integrate the equations of motion based on the Leapfrog algorithm [62]. Coordinate files were recorded every 1 ps. Neighbour searching was performed by the Verlet algorithm [63]. The Particle Mesh Ewald (PME) method, with a cut-off of 10 Å, was used to treat long-range electrostatic interactions at every time step. The van-der-Waals interactions were modelled using a 6–12 Lennard–Jones potential. The initial velocities were reassigned according to the Maxwell–Boltzmann distribution. For the relaxed models, relaxed apo-h and relaxed holo-h, to avoid the H-bonds formation, the following additional constraints were applied: the distances between the Cα-atoms from G62-Q78 and G60-N80 were maintained ≥ 9.9 Å to ensure that no H-bonds could form between the backbone of the glycine residues and their respective potential donor. During the first 100 ns, the constraints between G62-Q78 and G60-N80 were maintained and then removed for the next 400 ns to fully relax the systems. The cleaved L-loop was simulated for 0.5 µs as a fully unconstrained entity and for 5 µs as a polypeptide with the 100 kcal/mol Cartesian positional restrains of the backbone atoms (peptide N-Cα-C-O) from terminal residues R33 and N80 (cleaved restrained L-loop) to preserve the R33···N88 distance that was observed in crystallographic structures [16] and the de novo model [14].

#### 4.2.4. The Gaussian Accelerated Molecular Dynamic Simulations

To estimate the parameters needed for the Gaussian accelerated Molecular Dynamics (GaMD) simulation [18,64], 50-ns of cMD trajectories of hVKORC1 were used. Then, the 500-ns GaMD trajectories of the relaxed holo-h and relaxed apo-h were generated, using as starting conformations the cMD conformations of the respective forms taken at t = 50 ns. The boosting was applied of both total and dihedral potential energies. The boosting energy threshold was set as the maximal total potential energy calculated during the cMD. The coordinates were recorded every 1 ps.

### 4.3. Protein–Protein Docking

Protein–protein docking was performed with the HADDOCK2.4 web server (https://wenmr.science.uu.nl/haddock2.4/, accessed on 10 February 2022). HADDOCK (High Ambiguity Driven protein–protein DOCKing) [26] is a protein−protein docking approach based on available biochemical or biophysical information to drive the docking process. The docking protocol consists of several steps with user-defined input parameters. First, the topologies and coordinates files are generated for each molecule separately and merged to generate the starting models of the complex. Second, 10,000 structures were randomly sampled and subjected to the rigid body energy minimisation (it0). Third, the best 200 structures were selected, and a semi-flexible simulated annealing in torsion angle space was performed on them (it1). Finally, the obtained structures after the previous step were refined in Cartesian space with an explicit solvent (TIP3P)—a short molecular dynamics stage. After, the water-refined structures were clustered using a 7.5 Å RMSD cut-off and sorted according to the HADDOCK score. The maximum number of clusters was set to 10, and the minimal cluster size was set to 4. All other input parameters were kept default. To guide the docking, a set of ambiguous interactions restraints (AIRs), a pair of cysteine residues, C43 of hVKOR and C37 of PDI, was provided. Docking simulations were run with the same conformation of PDI (ligand protein) and with a set of different conformations of hVKOR (target protein) except for benchmark trials for which each protein, PDI and hVKORC1, was considered as ligand and target.

### 4.4. Data Analysis

***Standard analysis***. Unless stated otherwise, the data analysis was performed using CPPTRAJ 4.25.6 program [65] of AmberTools20 for MD conformations taken every 10 ps of simulation after least-square fitting [66,67] on the initial conformation (t = 0 ns) of a region of interest, thus removing rigid-body motion from the calculations.
(1)RMSD and RMSF values were calculated for the Cα-atoms using the initial model (at t = 0 ns) as a reference.(2)Secondary structural propensities for all residues were calculated using the Define Secondary Structure of Proteins (DSSP) method [68]. The secondary structure types were assigned for residues based on backbone − NH and − CO atom positions. Secondary structures were assigned every 10 and 20 ps for the individual and concatenated trajectories, respectively.(3)The trajectories of the cleaved L-loop of different forms were compared by means of best-fit RMSD values of all C-alpha atoms in a pairwise manner between two trajectories frame by frame. The resulting two-dimensional pairwise RMS2D matrix allows for finding the pairs of conformations with the minimal values of RMSD (less than the selected threshold values of 4 and 5 Å).(4)H-bonds between heavy atoms (*N*, *O*, and *S*) as potential donors/acceptors were calculated with the following geometric criteria: donor/acceptor distance cut-off was set to 3.6 Å, and the bond angle cut-off was set to 120°. Hydrophobic contacts were considered for all hydrophobic residues with side chains within 4 Å of each other.(5)The trajectories of the cleaved L-loop (holo-h and apo-h forms and de novo model) were compared by means of best-fit RMSD values of all C-alpha atoms in a pairwise manner between two trajectories frame by frame. The resulting two-dimensional pairwise RMSD matrix allows for finding the pairs of conformations with the minimal values of RMSD (less than the selected threshold values of 4 and 5 Å).(6)The radius of gyration $\left(Rg\right)$ was calculated from the atomic coordinates of the non-hydrogen atoms using the formula (Equation (1)) from [69]:(1)$$R_{g}=\sqrt{\frac{\sum_{i=1}^{N}m_{i}r_{i}^{2}}{\sum_{i=1}^{N}m_{i}}}$$
where $m_{i}$ is the mass of the atom i and $r_{i}$ is the distance of atom i from the centre of mass of the protein.


(1)The relative Gibbs free energy of the canonical ensemble was computed as a function of two reaction coordinates (Equation (2)) [70]:(2)$$\Delta G_{\left(R1,R2\right)}=-k_{B}T\ln\frac{P_{\left(R1,\;R2\right)}}{P_{max}}$$
where $k_{B}$ represents the Boltzmann constant, and *T* is the temperature. $P_{\left(R1,\;R2\right)}$ denotes the probability of states along the two reaction coordinates, which is calculated using a k-nearest neighbour scheme, and $P_{max}$ denotes the maximum probability.(2)The conformational landscape was reconstructed on the trajectories of the unconstrained (0.5 µs) and constrained (5 µs) (with the constraints applied on the N- and C-ends) simulations of the cleaved L-loop from holo-h and apo-h forms and the de novo model using pairs of metrics; RMSD, Rg, helical content, and principal components (PC1 and PC2) were obtained by the Principal Component Analysis (PCA. The three-dimensional representations of the free energy surface were plotted using MATLAB (US, © 2022–2021 The MathWorks, Inc., Natick, MA, USA).


#### 4.4.1. Advanced Data Analysis

To group the conformations with similar secondary structure patterns of the concatenated trajectory combined with four 5-µs cMD simulations of the cleaved L-loop (holo-h, apo-h, relaxed-apo-h, and de novo models), the algorithm of measuring the secondary structure similarity as described in [23] was used. The scoring matrix (secondary structure elements similarity matrix) supporting the 8-DSSP state alphabet was utilised. Then, the single-link hierarchical clustering method [46] with a threshold of 0.65 was carried out using SciPy (http.//https://scipy.org, accessed on 10 February 2022) to finally group similar conformations.

#### 4.4.2. Visualisation and Figure Preparation

Visual inspection of the conformations and figure preparation were performed with PyMOL (https://pymol.org/2/, accessed on 10 February 2022). To visualise the motions along the principal components, the Normal Mode Wizard (NMWiz) plugin [71], which is distributed with the VMD program, was used.

## Figures and Tables

**Figure 1 ijms-23-03899-f001:**
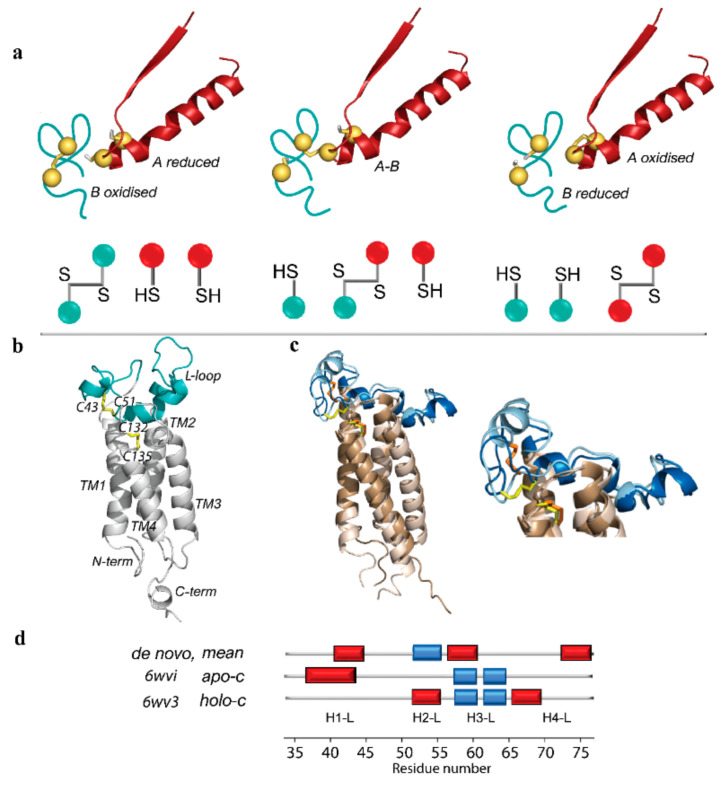
Thiol–disulfide exchange reactions between PDI and hVKORC1. (**a**) Thiol–disulfide exchange reactions involve reduced (proton-coupled) PDI (*A*, in red) and oxidised hVKORC1 (*B*, in deep teal). Generation of mixed-disulfide intermediate *A*-*B* involves an in-line transition state. Mechanistic models (top panel) and simplified scheme (bottom panel) of states *A*, *A*-*B,* and *B* show a reduction of pair of cysteine residues of oxidised protein (*A*, hVKORC1) by a CX_1_X_2_C dithiol motif (*B*, PDI). (**b**) Three-dimensional de novo models of hVKORC1 in the oxidised inactive state (left, [14]). (**c**) Superimposed crystallographic structures characterising the oxidised state of VKOR-like protein from *Takifugu rubripes* (free-ligand) (6wvi) and hVKORC1 with ligands (warfarin and glycerol monooleate) (6wv3), referenced as apo-c (light blue) and holo-c (dark blue) forms (middle), with a zoomed view on the L-loop (right). Ligands are not shown for clarity. (**b**,**c**) Protein and its fragment are shown as ribbons; disulfide bridges denoted in yellow and orange sticks. TMD helices coloured in grey (**b**) and brown (**c**). (**d**) Interpretation of L-loop secondary structure in the de novo model (mean conformation over cMD trajectory) and two crystallographic forms of oxidised VKOR. α- and 3_10_-helices are shown in red and blue, respectively. Helices numbering is adapted from [14].

**Figure 2 ijms-23-03899-f002:**
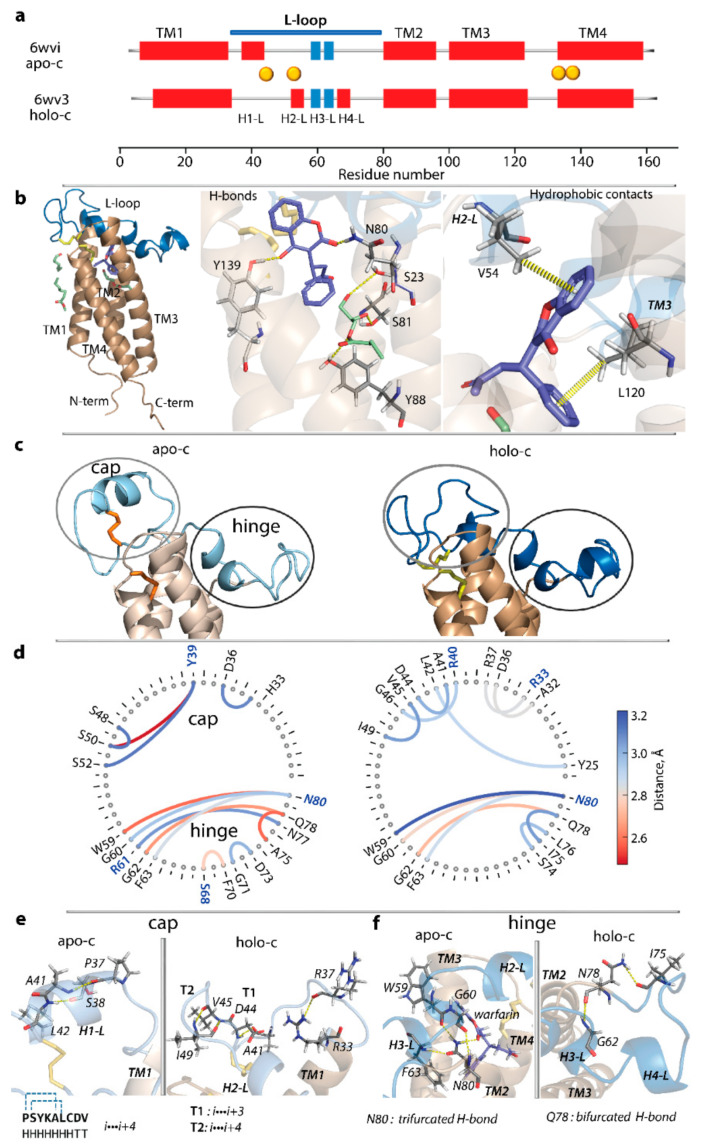
Structure of two crystallographic forms of the oxidised state of VKOR. (**a**) Secondary structures interpretation (DSSP) of crystallographic structures of VKOR-like protein from *Takifugu rubripes* (free ligand, 6wvi) and human VKOR (co-crystallised with warfarin and glycerol monooleate, 6wv3). α- and 3_10_-helices are shown in red and blue, respectively, and numbered as in [15]. L-loop segment is delimited by a blue bar; positions of cysteine residues are indicated by orange balls. (**b**) Structure of human VKORC1 co-crystallised with warfarin and glycerol monooleate (6wv3) (left). H-bonds (middle) and hydrophobic interactions (right) stabilising warfarin and glycerol monooleate in structure 6wv3. (**c**) L-loop conformation from crystallographic structures 6wvi (apo-c) and 6wv3 (holo-c). (**d**) H-bonds stabilising the cap and hinge in L-loop in two crystallographic forms showed as a string diagram. Residues contributing to H-bonding by their side chains are labelled in blue bold. (**e**,**f**) H-bonds stabilising α-helix and turns in L-loop cap in two crystallographic forms (**e**) and bifurcated and trifurcated H-bonds of residues N80 and Q78 (**f**) in both crystallographic structures, illustrated for holo-c form. (**b**,**c**,**e**) Protein is shown as cartoon with L-loop in blue (6wvi) and light blue (6wv3), TMD domain in brown (6wvi) and light brown (6wv3), and disulfide bridges as yellow (6wvi) and orange (6wv3) sticks; non-covalent interactions (H-bonds and hydrophobic contacts) are shown by yellow dashed lines. Figures were prepared from atomic coordinates. Numbering of residues corresponds to the respective sequences of crystallographic data.

**Figure 3 ijms-23-03899-f003:**
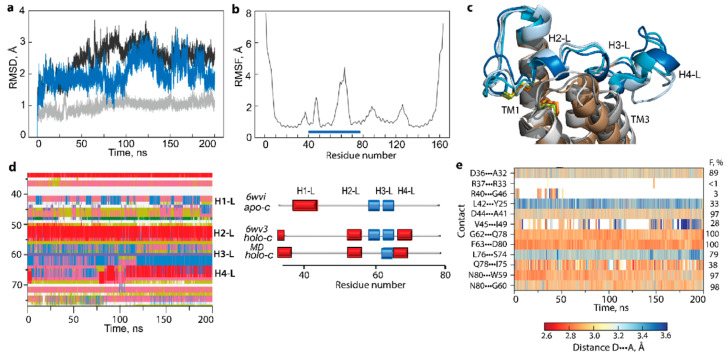
Analysis of cMD simulation of VKOR holo-c form without warfarin. **(a)** RMSDs from initial coordinates computed for the whole protein Cα-atoms (black), TMD (grey), and L-loop (blue) after fitting on the crystallographic structure-related domain. (**b**) RMSFs computed for Cα-atoms after fitting on the crystallographic structure. Blue horizontal bar delimits L-loop residues. (**c**) Superimposition of L-loop conformations randomly picked at t = 165 and 200 ns on crystallographic structure (6wv3). Protein is shown as cartoon with disulfide bridges as sticks. VKOR conformations and their structural elements are distinguished by colour: TMD helices, L-loop, and disulfide bridges in sand, dark blue, and yellow (6wv3); in dark grey, sky blue, and lemon (cMD frame picked at t = 165 ns); and in light grey, light blue, and orange (cMD frame picked at t = 200 ns). (**d**) (Left) Time-dependent evolution of each residue secondary structures as assigned by the Define Secondary Structure of Proteins (DSSP) method for L-loop: α-helices are in red, 3_10_-helices in blue, turn in orange, and bend in dark yellow. (Right) Interpretation of L-loop secondary structures in two crystallographic forms (two first lines) and for the average MD conformation (third line) of holo-c form. α- and 3_10_-helices are shown in red and blue, respectively, and numbered as in [15]. (**e**) Frequency (F, %) and length of H-bonds observed upon cMD simulation of holo-c form.

**Figure 4 ijms-23-03899-f004:**
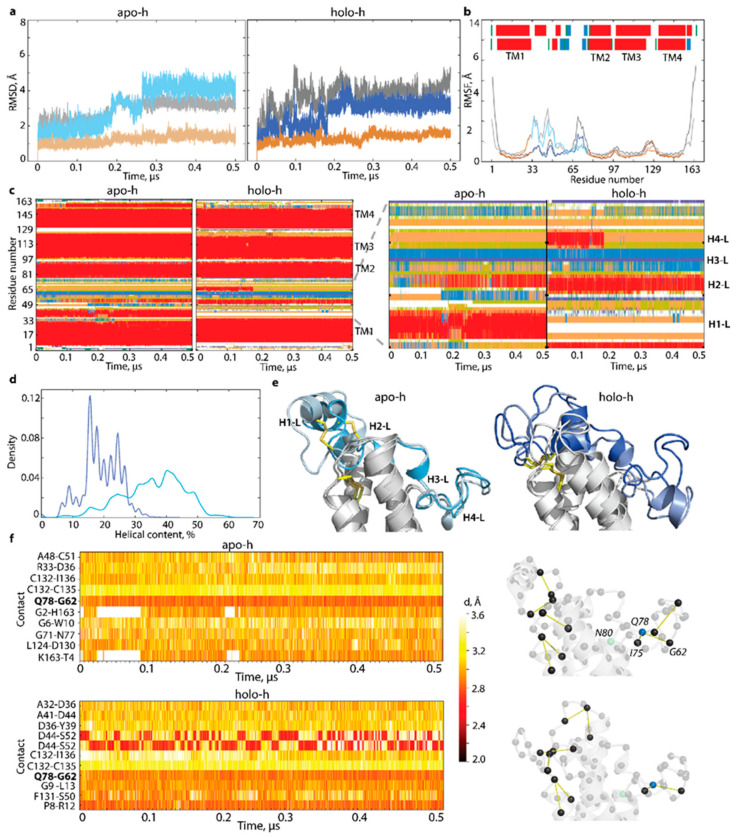
Accelerated MD simulations (GaMD) of human VKORC1 homology models, derived from crystallographic structures. (**a**) RMSDs computed for Cα-atoms of the full-length protein, TMD, and L-loop after fitting on the initial conformation (t = 0 ns) of the related domain. (**b**) RMSFs computed for the Cα atoms of two forms GaMD conformations after fitting on the respective average conformations. (**a**,**b**) Apo-h and holo-h forms are in light and dark colours, respectively: L-loop in blue, TMD in brown, total hVKORC1 in grey. (**c**) Time-dependent evolution of each residue secondary structures as assigned by DSSP method for L-loop: α-helices are in red, 3_10_-helices in blue, turn in orange, and bend in dark yellow. (**d**) Helical content of each form. (**e**) Superimposition of hVKORC1 conformations picked at 0 and 500 ns. Protein is shown as cartoon with disulfide bridges-forming cysteine residues in yellow sticks. TMD helices are in grey; L-loop conformations are distinguished by colour: t = 0 and 500 ns are in grey and blue for apo and dark blue and violet for holo-c form. (**f**) (Left) Non-covalent contacts (time series of H-bond events for H-bonds observed with frequency ≥ 0.8) in two forms of the oxidised VKOR. The contacts strength is shown by colour, from the strongest (2.7 Å, in red) to the weakest (3.6 Å, in white). (Right) Graphs of non-covalent contacts zoomed in on L-loop. Vertices represent residues, and a link between two residues reflects H-bonding.

**Figure 5 ijms-23-03899-f005:**
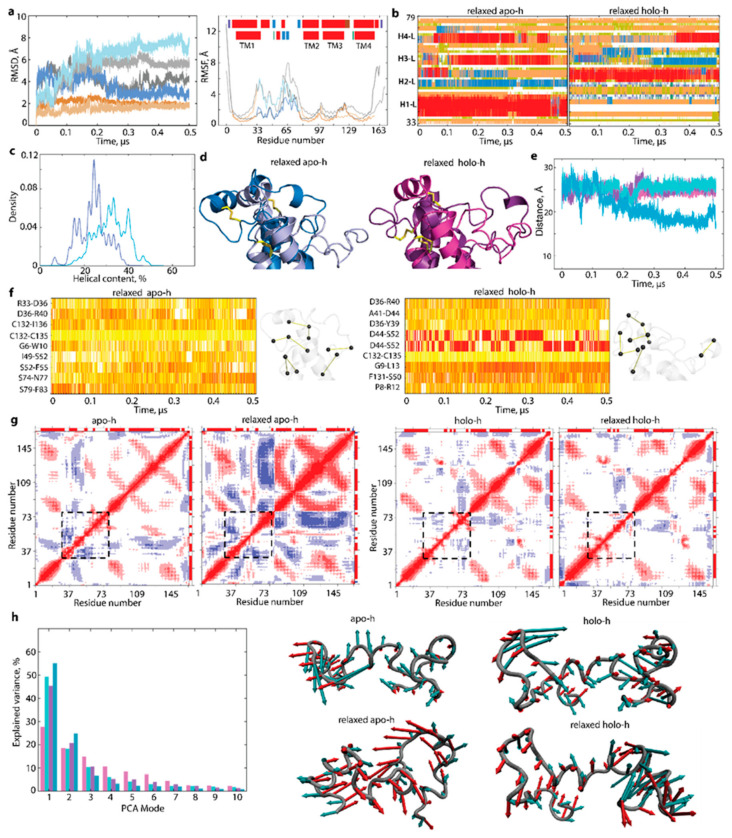
Conventional MD simulations of the homology models of hVKORC1 with conditioned H-bond Q78···G62. (**a**) RMSDs from the initial coordinates computed for Cα-atoms of the full-length protein (grey), TMD (brown), and L-loop (blue) after fitting to the initial conformation (t = 0 ns) of each related domain (left). RMSFs computed for the Cα atoms of each form after fitting on the initial conformation (right). (Insert) Secondary structures interpretation of each form’s average structure, relaxed apo-h (top), and relaxed holo-h (bottom). α- and 3_10_-helices are shown in red and blue, respectively. Conformations are distinguished by colour: relaxed holo-h and relaxed apo-h are in dark and light, respectively. (**b**) Time-dependent evolution of each residue secondary structures as assigned by DSSP for L-loop: α-helices are in red, 3_10_-helices in blue, turn in orange, and bend in dark yellow. (**c**) Helical content of each form. (**d**) Superimposition of randomly chosen conformations of relaxed apo-h and holo-h forms (in dark colours) into the conformations taken at t = 0 ns (in light colours). Protein is shown as cartoon with disulfide bridges-forming cysteine residues in yellow sticks. (**e**) Distance between the N- and C- ends of L-loop. (**f**) Non-covalent contacts (occurrence ≥ 80%) in two forms of the oxidised VKOR. The strength of contacts is shown by colour from the strongest (2.7 Å, in red) to the weakest (3.6 Å, in white). Graphs of each form of non-covalent contacts are shown on the right. Vertices represent residues, and a link between two residues reflects H-bonding. (**g**) Intrinsic motion of hVKORC1 and its L-loop represented by the inter-residue cross-correlation maps computed for Cα-atom pairs after fitting on the initial conformation of apo-h and holo-h forms and their relaxed versions. Secondary structures are projected onto the matrices borders (α-helix/β-strand in red/blue). L-loop is delimited with dashed lines. Correlated (positive) and anticorrelated (negative) motions between the Cα-atom pairs are shown as a red–blue gradient. (**h**) L-loop PCA modes calculated for each MD trajectory after least-square fitting of the MD conformations to the average conformation. The bar plot gives the eigenvalue spectra in descending order for the first 10 modes (left). Conformations are distinguished by colour: apo-h is in blue light, relaxed apo-h is in rose, holo-h is in dark blue, and relaxed holo-h is in purple (middle and right). Atomic components in the first PCA modes of the L-loop in apo (middle) and holo (right) form are drawn as red (1st mode) and blue (2nd mode) arrows projected onto the respective average structure. Only motion with an amplitude ≥ 4 Å is shown.

**Figure 6 ijms-23-03899-f006:**
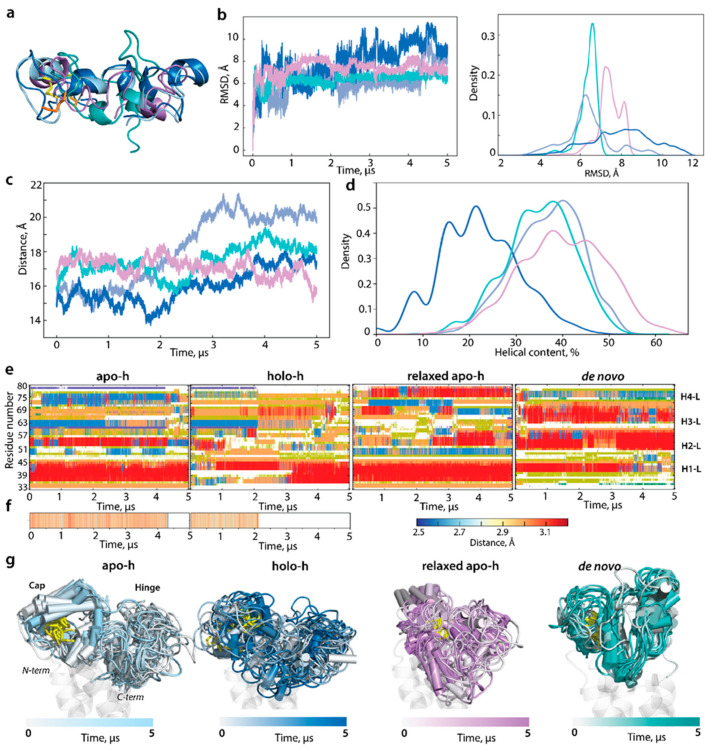
Structure and conformation of hVKORC1 L-loop studied as isolated polypeptide. (**a**) L-loop was extracted from apo-h and holo-h, relaxed apo-h, and de novo models and simulated (cMD) over 5 µs. (**b**) RMSDs of each form after Cα-atoms fitting to their respective initial conformations (left) and RMSD probability density (right). (**c**) Distance between L-loop ends (Cα−Cα atoms). (**d**) RMSFs computed for the Cα atoms of each form after fitting on their respective initial conformation. (**d**) Helical content of each model. (**e**) Time-dependent evolution of secondary structure in each L-loop form, (from left to right) apo-h, holo-h, relaxed apo, and de novo, as assigned by DSSP: α-helix is in red, 3_10_-helix is in blue, turn is in orange, and bend is in dark yellow. (**f**) Time-dependent evolution of H-bond Q78···G62 in L-loop cleaved from apo-h and holo-h forms. (**g**) Superimposition of 20 conformations taken each 25 ns of cleaved L-loop cMD simulations. The colour gradient shows the time-dependent conformations, from light (t = 0) to dark (t = 5 µs). (**a**,**g**) Protein is shown as cartoon with disulfide bridges in yellow sticks. L-loop cleaved from different hVKORC1 forms is distinguished by colour: blue light (apo-h), dark blue (holo-h), teal (de novo), and lilac (relaxed apo-h).

**Figure 7 ijms-23-03899-f007:**
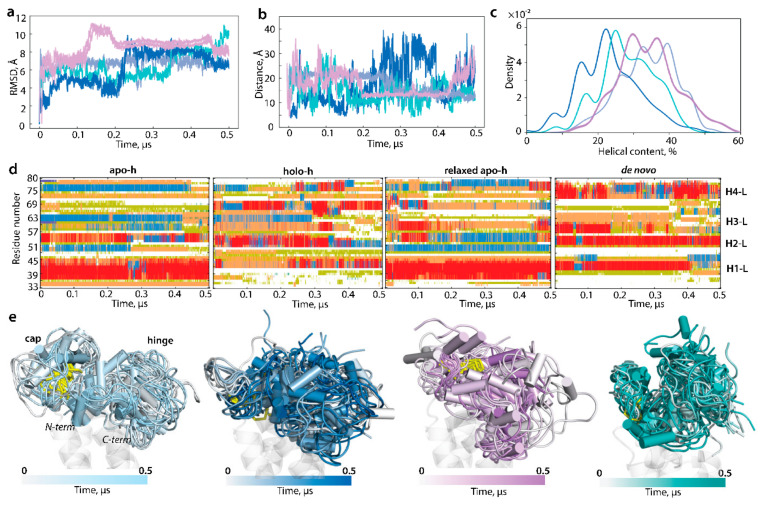
Structure of cleaved L-loop studied as fully released polypeptide. (**a**) RMSDs of each form after Cα-atoms fitting to their respective initial conformation. (**b**) Distance between L-loop ends (Cα−Cα atoms). (**c**) Overall helical content (left), the content of α- (middle), and 3_10_-helices (right) observed in cMD conformations of each form. (**d**) Time-dependent evolution of each L-loop form secondary structures (from left to right) apo-h, holo-h, relaxed apo, and de novo, as assigned by DSSP: α-helix is in red, 3_10_-helix is in blue, turn is in orange, and bend is in dark yellow. (**e**) Superimposition of 20 conformations taken each 25 ns of cMD simulations of L-loop relaxed apo-h and holo-h forms. The colour gradient shows the evolution of the trajectory, from light (t = 0) to dark (t = 5 µs). Protein is shown as cartoon and disulfide bridges as yellow sticks. The studied entities of L-loop are distinguished by colour: apo-h is in light blue, holo-h is in dark blue, relaxed apo-h is in lilac, and de novo model is in teal.

**Figure 8 ijms-23-03899-f008:**
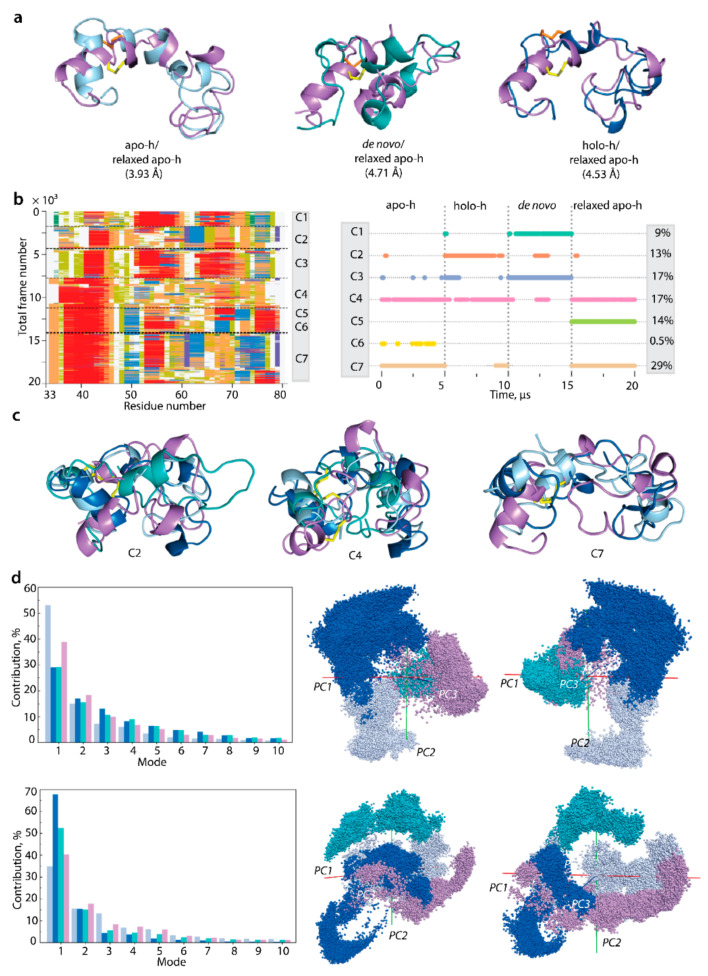
Clustering of MD conformations of L-loop simulated as an isolated polypeptide cleaved from each form—apo-h holo-h, relaxed apo-h, and de novo model. (**a**) Conformations with the lowest RMSD values (<4.0 and 5.0 Å, shown in parentheses) are superimposed. (**b**) Secondary structures-based clustering [23] with threshold 0.65 (left) performed on the time-depended evolution of each L-loop form secondary structures as assigned by DSSP: α-helix is in red, 3_10_-helix is in blue, turn is in orange, and bend is in dark yellow. Population and composition of clusters (right). The simulation time is displayed as cumulative for four trajectories. (**c**) Superimposition of L-loop from the most populated clusters C2, C4, and C7, composed of conformations generated from at least three different forms (**b**). (**d**) PCA analysis of concatenated cMD trajectories of cleaved L-loop simulated with constrained N- and C-ends (top panel) and without constraints (bottom panel). The bar plot gives the eigenvalue spectra in descending order for the first 10 modes calculated on each cMD trajectory (left). Projection of L-loop cMD conformations onto the first three principal components (PC) (right). The concatenated trajectories were least-square fitted on the mean conformation to remove rigid-body motions. (**a**–**d**) L-loop from apo-h (blue light), holo-h (blue dark), relaxed apo-h (lilac), and de novo (teal) is shown as cartoon with helices and disulfide bridges in yellow sticks.

**Figure 9 ijms-23-03899-f009:**
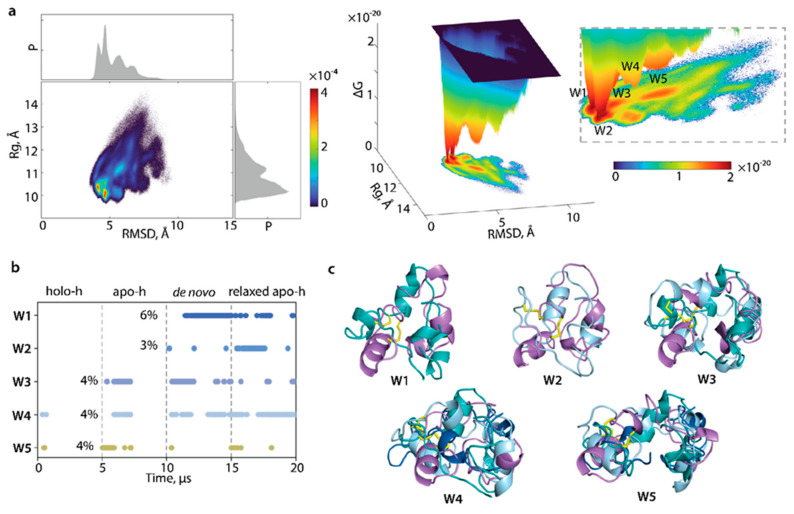
Free energy landscape $\left(FEL\right)$ of cleaved L-loop as a function of reaction coordinates, Rg versus RMSD. (**a**) Two-dimensional- (left) and three-dimensional (middle) representations of the L-loop conformational ensembles’ relative Gibbs free energy, zoomed in on the principal wells (right). Probability density (P) of each reaction coordinate is shown on the 2D map top and right. (**b**,**c**) Content of each well (W1–W5) illustrated as the superimposed representative conformations. All conformations were generated by cMD simulations of different forms of L-loop and fitted on the average conformation calculated on the merged data. The red colour on 2D and 3D diagrams represents the highest occurrence, yellow and green low, and blue represents the lowest occurrence. The free energy surface was plotted using MATLAB. L-loop is shown as a cartoon distinguished by colour: the holo-h is in light blue, the apo-h is in dark blue, relaxed apo-h is in lilac, and de novo model is in teal. Content of wells for the merged data is shown over the concatenated time range.

**Figure 10 ijms-23-03899-f010:**
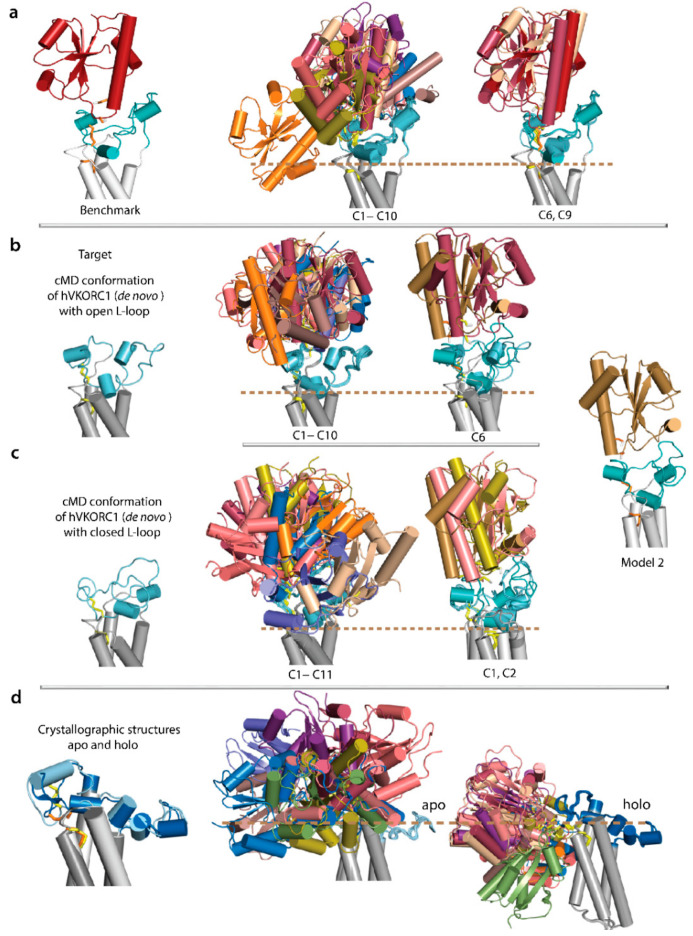
Computational protein–protein docking of PDI (ligand) onto hVKORC1 (target) performed with HADOCK using an information-driven method. (**a**) Benchmark complex from [15] (left); superimposition of the top 10 solutions (middle) and best solutions, clusters C6 and C9, (right) on benchmark. (**b**,**c**) (Left) hVKORC1 conformations with different L-loop shapes, elongated (open, **b**) and compact (closed, **c**) conformations in de novo model [14]. (Middle) Superimposition of the top solutions obtained for hVKORC1 with closed (**b**) and open (**c**) L-loop conformations. (Right) Model 2 suggested as possible in [15] used for superimposition of HADDOCK solutions, cluster C6, and C1 and C2 obtained for hVKORC1 with open (**b**) and closed (**c**) L-loop conformations. Two orthogonal projections are shown. (**d**) hVKORC1 homology models, apo-h, and holo-h, quasi-identical to X-ray structures (PDB IDs: 6wv3 and 6wvi) (left). Superimposition of the top of 10 HADDOCK solutions for PDI docked onto apo-h (middle) and holo-h (right) forms of hVKOR. (**a**–**d**) Protein is shown as a cartoon with helices as cylinders and disulfide bridges in yellow sticks. A possible boundary of the membrane is denoted as dashed line.

**Figure 11 ijms-23-03899-f011:**
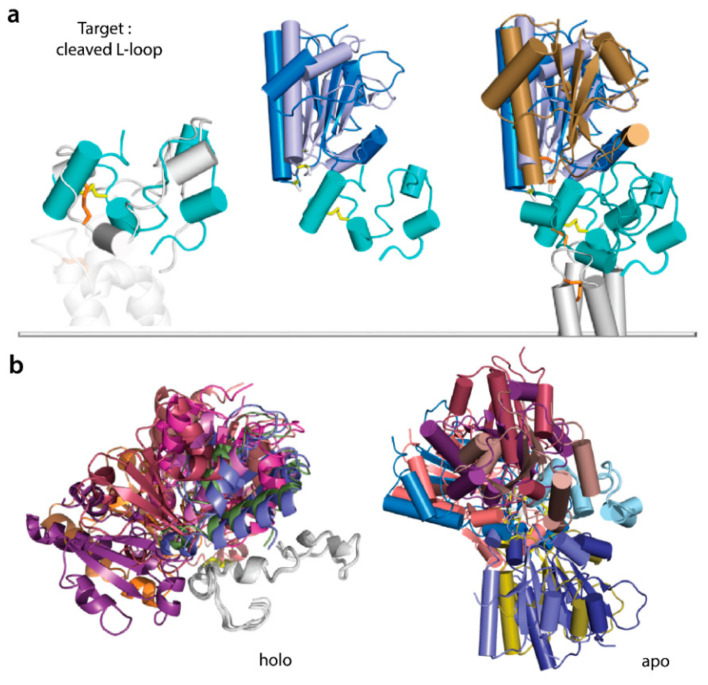
Protein–protein computational docking of PDI (ligand) constant conformation and cleaved L-loop (target) represented by different conformations. Docking was performed by HADDOCK using an information-driven method. (**a**) (left) Superimposition of L-loop cleaved from Model 1 of PDI–hVKORC1 complex at t = 441 ns of cMD simulation (grey) [15] and taken at 4.62 µs of cMD simulation of cleaved L-loop from apo-h form simulated as an isolated species (teal). (Middle and right) The two best docking solutions obtained for PDI docking onto L-loop conformation with compact (closed) shape superimposed onto L-loop from Model 2 (middle) and on Model 2 (right), suggested as possible in [15]. (**b**) Superimposition of the best solutions obtained for PDI docking onto cleaved L-loop from holo- and apo-forms (X-ray). (**a**,**b**) Protein is shown as a cartoon with helices as cylinders or spiral and disulfide bridges as yellow sticks.

## Data Availability

The numerical model simulations, upon which this study is based, are too large to archive or transfer. Instead, we provide all the information needed to replicate the simulations. The model coordinates are available from L. Tchertanov at ENS Paris-Saclay.

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
