# Peer review of "Human Vitamin K Epoxide Reductase as a Target of Its Redox Protein"

_ijms, 2022, doi:10.3390/ijms23073899_

Round 1

Reviewer 1 Report

General comment:

This manuscript, entitled “Human Vitamin K Epoxide Reductase as a target of its Redox Protein,” authored by Ledoux et al., reported the folding and conformational plasticity of Human vitamin K epoxide reductase (hVKORC1) in inactive form using molecular dynamics (MD) simulation. The docking study of protein disulfide isomerase (PDI) onto different hVKORC1A reflects the protein-protein interaction through conformation selection of L loop in the oxidized state. This study will provide essential links between protein-redox partner interaction and electron delivery for its enzymatic activity. This kind of bioinformatic approach is beneficial for exploring membrane proteins as expression and purification are always challenging task. In my opinion, this is a valuable work and is suitable for publication in the International Journal of Molecular Sciences after the authors have addressed the following comments and questions:

Specific comments:

  • Line 88 – what are the four different thioredoxin proteins tested for potential redox partners? And what is the basis of selecting PDI?
  • Is L-loop is the sole region for protein-protein interaction?
  • What kind of interaction is predominant here – hydrophobic or electrostatic?
  • Is MD simulation able to track electron transfer path? How beta-hairpin and loop 3-4 is involved?

Author Response

Responses to Reviewer #1:

This manuscript, entitled “Human Vitamin K Epoxide Reductase as a target of its Redox Protein,” authored by Ledoux et al., reported the folding and conformational plasticity of Human vitamin K epoxide reductase (hVKORC1) in inactive form using molecular dynamics (MD) simulation. The docking study of protein disulfide isomerase (PDI) onto different hVKORC1A reflects the protein-protein interaction through conformation selection of L loop in the oxidized state. This study will provide essential links between protein-redox partner interaction and electron delivery for its enzymatic activity. This kind of bioinformatic approach is beneficial for exploring membrane proteins as expression and purification are always challenging task. In my opinion, this is a valuable work and is suitable for publication in the International Journal of Molecular Sciences after the authors have addressed the following comments and questions:

Response: The Authors thank Reviewer 1 for his positive comments on our manuscript. Our responses for specific comments are supplied after each concern.

Specific comments

  • Line 88 – what are the four different thioredoxin proteins tested for potential redox partners? And what is the basis of selecting PDI?

Response: We previously studied structural and dynamical features of four thioredoxins, PDI, Erp18, TMX1 and TMX4, suggested as the most probable redox partners for hVKORC1 (https://dx.doi.org/10.1073/pnas.1009972107), and we concluded that PDI is the most suitable hVKORC1 partner (Stolyarchuk M.; Ledoux, J.; Maignant, E.; Trouvé, A.; Tchertanov, L. Identification of the Primary Factors Determining the Specificity of Human VKORC1 Recognition by Thioredoxin-Fold Proteins. Int. J. Mol. Sci. 2021, 22, 802. doi: 10.3390/ijms22020802). Two PDI segments were found to be the most likely interacting fragments with hVKORC1. One of them, showing partial intrinsic disorder, was considered the most compatible with hVKORC1’s disordered L-Loop to molecular complex formation through their mutual adaptability in the recognition/binding process. This choice was supported by numerous empirical and theoretical studies of the disordered  fragments role in recognition/formation of molecular complexes (https://dx.doi.org/10.1016/j.jmb.2006.07.087). As VKORC1-protein interactions were mainly observed under co-immunoprecipitation assays with several proteins, specific partner(s) is yet to be empirically identified, to this day, no structural data of VKOR complex with its partners are available. A little further in advance, I would like to mention that our identification of PDI as a partner of hVKORC1 has already been confirmed experimentally (in vitro) by biologists (private communication).

  • Is L-loop is the sole region for protein-protein interaction?

Response: As hVKORC1 is composed of stable transmembrane domain and flexible and disordered luminal loop (L-loop) protruded on the ER-membrane surface, we considered that L-loop is the most suitable region for protein-protein interaction. This suggestion is based on structure and functions of well-studied bacterial VKOR homologue.      

  • What kind of interaction is predominant here – hydrophobic or electrostatic?

Response: We have addressed this question in (Stolyarchuk et al. 2021, doi: 10.3390/ijms22020802) where we analysed the intermolecular contact network between hVKORC1 and differently oriented PDI (Model 1 and Model 2). We found that the protein-protein binding is provided by salt bridges, strong hydrogen bonds and hydrophobic contacts (see insert).

  • Is MD simulation able to track electron transfer path? How beta-hairpin and loop 3-4 is involved?

Response: The electron transfer is not the subject of this paper. Moreover, classical or accelerated methods of MD simulation, used in the present study, are not appropriate for this type of investigations. To probe the thiol-disulfide exchange reaction between hVKORC1 and its partner, hybrid QM/MM calculations should be performed. The models of PDI-hVKORC1 complex either proposed in (Stolyarchuk et al. 202, doi:10.3390/ijms22020802), or reported in this paper (docking models) can be used as initial models in QM/MM simulation. In these models, the region located between Cys43—Cys51 and bordering the HL-1 helix, called beta-hairpin, is unstructured and mobile. This region is thought to be involved in the transport of Cys43–Cys51 close to Cys132—Cys135 (active site) located in the loop 3-4 region, between the transmembrane helices TM3 and TM4.

Reviewer 2 Report

Fantastic paper, koagulation in introduction can also be linked to German. Methodology on a very high level, Intriguiging results. The only thing I miss is more specific discussion of genome effects on the different proteins involved in this study. Also PIVKA-II has been used to define a subclinical K-vitamin defect (normal PT-values). Can the authors elaborate on deficits in vitamin K or drug effects making the vitamin K-cycle dysfunctional (eg. through the different mechanisms on making VKOR functional)?

Results, figures and discussion on very high level.

Author Response

Responses to Reviewer #2:

  • Fantastic paper, koagulation in introduction can also be linked to German. Methodology on a very high level, Intriguiging results... Results, figures and discussion on very high level.

Response: Thank you so much for your positive comments of the manuscript. We really appreciate you taking the time to share your experience with us — and we agree that coagulation is a German term. Henrik Dam referred to both “Scandinavian and German spelling” of the word koagulation in his Nobel Prize lecture (https://www.nobelprize.org/uploads/2018/06/dam-lecture.pdf).

Regarding your very positive comments, we, the whole team, are very happy. Many many thanks… We are working every moment to obtain a very positive response to our research, and we are glad to hear that we have achieved that goal, at least from your point of view, and I hope that the reaction of other future readers will be the same.

  • The only thing I miss is more specific discussion of genome effects on the different proteins involved in this study.

Response: This study is conducted regardless of over/under expression of VKORC1. As of today, VKORC1 is not suggested as a signalling protein, and potential cytoplasmic partners through direct or indirect interaction are not reported. Its function regarding any control of gene expression is not reported either.

  • Also PIVKA-II has been used to define a subclinical K-vitamin defect (normal PT-values). Can the authors elaborate on deficits in vitamin K or drug effects making the vitamin K-cycle dysfunctional (eg. through the different mechanisms on making VKOR functional)?

Response: Protein Induced by Vitamin K Absence II and its role in vitamin K regulation/transformation is a very special question and requires a careful study.  
